# Exact Thermodynamics and Transport in the Classical Sine-Gordon Model

R. Koch[1*], A. Bastianello[2, 3]

**1** Institute for Theoretical Physics, University of Amsterdam, PO Box 94485, 1090 GL Amsterdam, The Netherlands
**2** Department of Physics and Institute for Advanced Study, Technical University of Munich, 85748 Garching, Germany
**3** Munich Center for Quantum Science and Technology (MCQST), Schellingstr. 4, D-80799 München, Germany
\* r.koch@uva.nl

April 11, 2023

## Abstract

We revisit the exact thermodynamic description of the classical sine-Gordon field theory, a notorious integrable model. We found that existing results in the literature based on the soliton-gas picture did not correctly take into account light, but extended, solitons and thus led to incorrect results. This issue is regularized upon requantization: we derive the correct thermodynamics by taking the semiclassical limit of the quantum model. Our results are then extended to transport settings by means of Generalized Hydrodynamics.

# 1  Introduction

The sine-Gordon model is a one-dimensional relativistic field theory appearing in the most diverse contexts. For example, it describes the low energy physics of a plethora of systems, ranging from spin chains [1–4], spinful atoms [5], arrays of Josephson's junctions [6,7], certain quantum circuits [8,9] and weakly tunneled-coupled quasicondensates [10–15]. Its ubiquity is closely related to the vast applicability of bosonization [5, 16], since sine-Gordon can be regarded as one of the most natural massive perturbations of a gapless Luttinger-Liquid. Consequently, this model is well known and widely studied by a broad community, both numerically and analytically: one of the salient features of this field theory is its integrability [17–19], which allows for a variety of analytic results on the one hand, and guarantees atypical thermalization [20] and transport [21, 22] on the other.

The versatility of sine-Gordon is further reflected in the multitude of angles from which it has been analysed, that closely follow the development of new experimental platforms. For example, deep mathematical questions in partial differential equations (PDEs) greatly fueled the development of the inverse scattering method for integrable differential equations [18,19], a category to which the classical sine-Gordon model belongs to. With the experimental progresses in manipulating quantum matter, the general interest shifted to explore the quantum integrability of sine-Gordon, first at the level of the few-body problem by determining the exact spectrum and scattering processes [23], then undertaking the ambitious form-factor bootstrap program to determine observables' matrix elements on scattering states [17]. In the meanwhile, new experiments [24–27] and numerical studies mimicking plausible experimental setups [11,15], showed that the sine-Gordon mass spectrum and form factors can be realized with great accuracy.

These last achievements fall under the umbrella of few-body excitations, but new advances in quantum simulators [28] are responsible for a further shift in focus: how to describe *macroscopically excited* states, both in and out of equilibrium?

While we have so far kept the quantum and classical sine-Gordon models on roughly the same footing, this last question represents a historic bifurcation between the two: at the quantum level, finite-temperature thermal states can be exactly characterized by means of the Thermodynamic Bethe Ansatz (TBA) [29]. These general ideas can also be applied to out-of-equilibrium sudden quantum quenches [30,31] and, more recently, to study transport within the framework of Generalized Hydrodynamics [22,32,33]. In contrast, a parallel development in the classical realm stands on far more shaking foundations.

Before embarking on a more detailed discussion and substantiating our last assertion, we wish to put forward the main goal of our work: in this paper, we study the thermody-

namics and hydrodynamic transport of the classical sine-Gordon model. Our motivations are multifaceted: while many works have been devoted to study the classical sine-Gordon thermodynamics with a variety of approaches [34–39], all of them led to inconsistent results [40] for a very good reason that will be clarified later on. Therefore, our work solves the long-standing problem of deriving the exact thermodynamics of the classical sine-Gordon field theory.

On a more pragmatic level, experimental realizations of sine-Gordon are often close to the classical limit. This is for example the case, when sine-Gordon emerges as the low-energy description of tunnel-coupled and weakly-interacting quasicondensates [10,11]: in the regime of a large number of atoms and weak interactions commonly realized in experiments [12], semiclassical methods account well for experimental observations. Therefore, our findings are relevant for experimental applications.

Lastly, our results will serve as a stepping stone for future developments in the quantum realm as well: with the fast development of new experimental techniques, the realization of a versatile sine-Gordon simulator deep into the quantum regime can be foreseen in the near future [15,41]. This calls for new theoretical advances able to keep the pace with the experimental progress: here, Generalized Hydrodynamics is a prominent candidate for taking on this ambitious challenge [42–45], but its application to sine-Gordon is largely untapped. So far, only transport in clean and homogeneous settings has been considered [46], but taking into account inhomoegeities present in realistic experimental setups represents a challenge on a whole new level [47–49]. In this perspective, developing first the Generalized Hydrodynamics of the classical sine-Gordon suitable for experimental configurations will be an irreplaceable laboratory to benchmark hydrodynamic ideas against ab-initio Monte Carlo simulations [50, 51], serving as a backbone for future Generalized Hydrodynamics results of the quantum model beyond reach of numerical checks. The present work is the first step in this long term program.

Our paper is structured as follows: We begin by introducing the classical sine-Gordon model in Section 2. We will comment on the previous strategies in the derivation of its exact thermodynamics, their inconsistencies and pitfalls. We will identify the presence of extended solitons with arbitrary small mass as the problem at the common root of these methods (see also Refs. [52,53] for related considerations). The crucial point comes from the coalescence of two competing effects: on the one hand, light excitations are highly excited at any finite temperature. On the other hand, any arbitrary large, yet finite, volume poses a cutoff on the maximum width of the allowed solitons.

A correct derivation of the thermodynamics passes through a regularization of these extended modes: while a solution of this riddle by means of purely classical considerations is highly desirable, it is still beyond our current understanding. To circumvent this issue, we take another route: in Section 3 we derive the classical thermodynamics by taking the semiclassical limit of the quantum model. While it may seem a complicated detour, quantization introduces a finite mass gap in the model and cures the aforementioned problem. Our final result shows that the sine-Gordon thermodynamics is fully describable by a collection of soliton-like excitations with a renormalized statistics. We wish to point out that we have identified a similar feature in a previous investigation on nonequilibrium states in the attractive Non-Linear-Schrödinger equation (NLS) [53]. The validity of our finding is assured by an analytic analysis of the low-temperature regime, which was not correctly captured by previous methods, and by comparison with ab-initio numerical simulations for arbitrary temperatures and couplings. After having dealt with equilibrium thermodynamics, in Section 4 we revert to nonequilibrium settings. In particular, we use Generalized Hydrodynamics to study parti-

tioning protocols [54] and observe transport: the agreement with numerical data is excellent. We gather our conclusions in Section 5 and provide an outlook on future directions stemming from our present findings. Some appendices discussing more technical aspects follow.

## 2 The classical sine-Gordon model

Sine-Gordon is a one-dimensional relativistic field theory governed by the following Hamiltonian

$$H = \int dx \frac{1}{2} c^2 \Pi^2 + \frac{1}{2} (\partial_x \phi)^2 + \frac{m^2 c^2}{g^2} (1 - \cos(g\phi)). \tag{1}$$

$H$ can be interpreted both on classical or quantum grounds. The only difference is in regarding the conjugated fields $\phi$ and $\Pi$ either as classical fields, thus obeying canonical Poisson brackets $\{\phi(x), \Pi(y)\} = \delta(x - y)$, or as operators with canonical commutators. Within this section, we focus solely on the classical case, while in Section 3 we will take a short detour into the quantum world. Above, $c$ is the light velocity, $m$ the bare mass scale and $g$ sets the interaction strength. In the classical case, all these quantities can be set to unity by a proper renormalization of fields, distances and overall energy scale. Nonetheless, we retain these couplings explicitly in view of the forthcoming semiclassical limit.

We begin with characterizing the excitations' content of the model [19, 55]. As it is self-evident from the periodic cosine potential, the sine-Gordon model has infinitely many degenerate ground states, or vacua, $\phi(x) = 2\pi n/g$ with $n \in \mathbb{Z}$. This degeneracy allows for the presence of topological excitations interpolating between neighboring vacua: these are called kinks or antikinks depending on whether the phase slip is positive or negative as $x$ is increased. The spatial profile of a soliton at rest is easily obtained by solving the classical equation of motion with unbalanced boundary conditions, resulting in [55] $\phi_K(x) = \frac{4}{g} \arctan(e^{-mcx}) + 2\pi/g$. The soliton configuration can be translated and set in motion by using relativistic invariance and boosting the spatial coordinate $\phi_K(x) \to \phi_{K,\theta}(t, x - x_0) = \phi_K(\cosh\theta(x - x_0) - \sinh\theta ct)$, where $\theta$ is the rapidity and $x_0$ the soliton position at $t = 0$. The antikink configuration is simply the reflected profile $\phi_{\bar{K},\theta}(t, x) = -\phi_{K,\theta}(t, x)$.

Since the kink's field configuration departs from the ground state, it has finite energy. In particular, as it is easy to check from the explicit solution $\phi_K$, kinks behave as relativistic particles with dispersion $\epsilon_K(\theta) = Mc^2 \cosh\theta$ where the soliton mass is

$$M = \frac{8m}{cg^2}. \tag{2}$$

The presence of topological excitations is a feature that is shared with many other non-integrable models, for example the $\phi^4$−field theory in a double well. In contrast, the peculiarity of sine-Gordon as an integrable field theory is manifested in the scattering events: due to the presence of infinitely many conservation laws, scattering is largely constrained and is non-diffractive [23]. Indeed, exact solutions to the equation of motion describing multi-kink states can be explicitly built through the inverse scattering method [18, 19]. For example, a two-soliton solution can be found [55] as

$$\phi_{K,\bar{K}}(t, x) = -\frac{4}{g} \arctan\left(\frac{\sinh(mc^2 t \sinh\theta)}{\tanh\theta \cosh(mcx \cosh\theta)}\right). \tag{3}$$

Notably, one has the limiting cases (assuming $\theta > 0$)

$$
\phi_{K,\bar{K}}(t,x) = \begin{cases} \phi_{K,\theta}\left(t, x - \frac{\varphi(2\theta)}{2Mc\cosh\theta}\right) + \phi_{\bar{K},-\theta}\left(t, x + \frac{\varphi(2\theta)}{2Mc\cosh\theta}\right) & t \to -\infty \\ \phi_{K,\theta}\left(t, x + \frac{\varphi(2\theta)}{2Mc\cosh\theta}\right) + \phi_{\bar{K},-\theta}\left(t, x - \frac{\varphi(2\theta)}{2Mc\cosh\theta}\right) - 2\pi/g & t \to +\infty \end{cases}
$$

$$(4)$$

Hence, Eq. (3) describes the scattering event of an incoming kink-antikink pair. Thanks to integrability, the scattering is completely elastic and the kink passes through the antikink without transferring energy to other modes. The effect of interactions manifests in a leap forward of the solitons after scattering quantified by the classical scattering shift

$$
\varphi(\theta) = \frac{8}{cg^2} \log\left(\frac{\cosh\theta + 1}{\cosh\theta - 1}\right) . \tag{5}
$$

Similarly, the kink-kink scattering shift can be derived and is identical to the kink-antikink one. Kinks and antikinks alone do not exhaust all the possible excitations: together with topologically-charged quasiparticles, neutral excitations are also present. These are called breathers and are parametrized by a real spectral parameter $\sigma \in [0,1]$. Breathers can be seen as kink-antikink boundstates: indeed, the kink-antikink scattering state (3) can be analytically continued to imaginary rapidities $\theta \to i\frac{\pi}{2}(1-\sigma)$ and still remains a solution of the equation of motion, even though it completely changes its character. After analytic continuation, the field configuration remains localized around $x = 0$ with a width $\ell_{\text{breather}} \sim 2/[mc\cos(\frac{\pi}{2}(1-\sigma))]$, but retains a non-trivial time dependence with a pulsating motion: this is a breather at rest. The rest energy of the new particle is readily obtained by analytic continuation of the two kinks energy $2Mc^2\cosh(\theta) \to 2Mc^2\sin(\frac{\pi}{2}\sigma)$. One therefore finds the breather masses as

$$
m_\sigma = 2M\sin\left(\frac{\pi}{2}\sigma\right) . \tag{6}
$$

Of course, likewise solitons, breathers can be set in motion by a Lorentzian boost and they are found to obey a relativistic dispersion law. After having identified these new excitations, the next task is to find their scattering properties. Fortunately, no new calculations are needed and the breather-kink and breather-breather scattering shifts can be derived from Eq. (5) through analytic continuation. First, one builds on the fact that, thanks to integrability, the scattering shifts of multi-particles behave additively [29]. For example, a kink with rapidity $\theta$ that collides with a kink and antikink of rapidities $\theta_a$ and $\theta_b$ experiences a phase shift $\varphi(\theta - \theta_a) + \varphi(\theta - \theta_b)$. Then, the scattering shift of a kink with rapidity $\theta$ and a breather with rapidity $\theta'$ is obtained by analytically continuing $\theta_a \to \theta' + i\frac{\pi}{4}(1-\sigma)$ and $\theta_b \to \theta' - i\frac{\pi}{4}(1-\sigma)$. Similarly, the breather-breather scattering shift is also computed

$$
\varphi_{\sigma,\sigma'}(\theta) = \frac{16}{cg^2} \log\left(\frac{[\cosh(\theta) - \cos((\sigma+\sigma')\pi/2)][\cosh(\theta) + \cos((\sigma-\sigma')\pi/2)]}{[\cosh(\theta) - \cos((\sigma-\sigma')\pi/2)][\cosh(\theta) + \cos((\sigma+\sigma')\pi/2)]}\right) . \tag{7}
$$

Consistently, it holds that $\lim_{\sigma' \to 1} \varphi_{\sigma,\sigma'}(\theta) = 2\varphi_\sigma(\theta)$ and $\lim_{\sigma \to 1} \varphi_\sigma(\theta) = 2\varphi(\theta)$, where $\varphi_\sigma(\theta)$ is the breather-kink scattering shift. Notice the simple normalization (see Appendix A)

$$
\int d\theta\, \varphi_{\sigma,\sigma'}(\theta) = \frac{32\pi^2}{cg^2} \min(\sigma, \sigma') . \tag{8}
$$

One could wonder if kinks, antikinks and breathers are a complete set of excitations for sine-Gordon: an inverse scattering analysis on the infinite system shows that these are the

only possible excitations of solitonic type [19], but it still leaves room for dispersive radiative modes. A priori, it is unclear if and how these modes should be taken into account. Slightly anticipating on the content of Section 3, we found that radiative modes do not contribute to the thermodynamic description of sine-Gordon as new entities distinct from solitons. In contrast, radiation can be viewed as a condensation of light breathers, as it will be unveiled by studying the low temperature regime.

## 2.1 Classical methods for thermodynamics and the large-solitons problem

Before moving to the core of our paper, we briefly overview the different approaches to the sine-Gordon's thermodynamics and their difficulties, identifying the common plague to these methods and outlining our strategy. We do not aim to give a comprehensive overview, but rather only point out some key observations: for a more extensive discussion, the reader can refer to the cited literature.

As we have already outlined, the first puzzle consists in identifying which are the relevant excitations for macroscopically excited (e.g. thermal) states: an inverse scattering analysis on the infinite system points at two radically different modes [19], namely solitons and radiation. On the one hand, non-dispersive solitonic modes are expected to obey a Maxwell-Boltzmann's type of statistics, similarly to other classical PDEs such as the Korteweg–De Vries (KdV) PDE [56]. In contrast, radiation should obey a Rayleigh-Jeans distribution, as it has been shown in classical PDEs with only radiative modes, such as sinh-Gordon [57, 58] or the defocusing Non-Linear-Schrödinger equation [59]. If, and how, these modes contribute to thermodynamics remained unclear so far. The story gets even more complicated if one attempts to rigorously take the thermodynamic limit: in the inverse scattering at finite volume, excitations are "quantized" through the complicated inverse gap solution of the transfer matrix [60,61] (reminiscent of the finite-volume Bethe-Ansatz equations of quantum systems [29]), and furthermore, there is no clear distinction between solitons and radiation anymore. In contrast, all excitations seem to have a solitonic flavor at finite volume: up to our knowledge, the effect of taking the thermodynamic limit has not been well-understood. Some simplifications can be made by neglecting the fine-structure of the finite gap solutions and approximating them in a coarse grain manner, by taking a continuum limit. Within this framework, the interaction-renormalized phase space density of models with a single excitation species can be recovered. See for example the sinh-Gordon model where only a radiative mode [57] is present. However, one may wonder if this approach can be replicated for models with more particle species or bound states thereof: in analogy with the string-charge duality of quantum models [62], higher ranks representation of the transfer matrix may be needed. At the best of our knowledge, the fulfillment of this program remains an open challenge in sine-Gordon.

An attempt to circumvent these technical bottlenecks has been made through more phenomenological approaches, primarily within the soliton-gas picture [61,63–65]. In this framework, one builds the thermodynamics of a gas of particles (the solitons) where interactions are kept into account by giving particles an effective finite length, matched with the soliton-soliton scattering shift. Radiation is entirely neglected. Soliton gases give quantitatively correct results for other solitonic models, such KdV [33, 61], but fails for sine-Gordon. Indeed, the so-derived thermodynamics lacks the correct low-temperature limit [40], where sine-Gordon must be well approximated by the non-interacting massive Klein-Gordon model. In passing, we stress that low temperatures excite only long wavelengths and, in contrast with sine-Gordon, the Klein-Gordon model features only radiation. Actually, both the attempts made

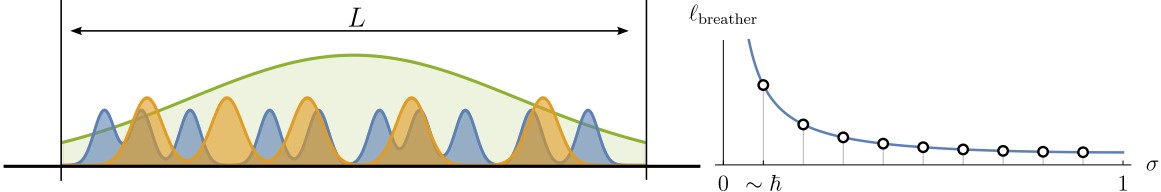

Figure 1: **Pictorial representation of the soliton gas picture and the role of extended solitons.** Left: In the classical field theory, breathers with arbitrary large extent exist, but on the other hand they cannot be placed in a finite, albeit large, volume $L$. Right: in the quantum sine-Gordon model, the spectral parameter is quantized $\sim \hbar$, putting a maximum cap on the soliton extension, thus acting as a regulator.

through the inverse scattering and the soliton-gas picture led to the same, albeit incorrect, set of integral equations describing the thermal states. We do not report them here, but they will be discussed once we have derived our result.

It should be apparent by now that both these methods suffer from putting the system in a finite volume of size $L$ and eventually taking $L \to \infty$, while retaining a finite excitation density. Within the soliton-gas picture, the problem is clear: in the infinite volume we find breathers whose mass and size are decided by the spectral parameter $\sigma$. As $\sigma$ approaches zero, their mass (6) vanishes and are therefore likely to be excited by thermal flucutations. One can expect these modes to be more and more relevant for $\sigma \to 0$. On the other hand, their spatial extension grows and any finite volume will act as a cutoff on the allowed breather $\ell_{\text{breather}} < L$, as we depict in Fig. 1. While the problem is clear, its solution is not evident and how to include the proper regularization in the soliton-gas picture remained elusive for us.

Therefore, we will now take a different root and use as a starting point the quantum sine-Gordon and its thermodynamics. As we will see, in the quantum model the breathers' spectral parameter is quantized in units of $\hbar$, acting as a cutoff on the maximum size of the breathers. Hence, by taking first the thermodynamic limit and only then the semiclassical limit, we converge to the correct result.

# 3   From the quantum to the classical thermodynamics

Semiclassical methods date back to the birth of quantum mechanics in a variety of contexts. Covering a comprehensive overview is impossible, neither the focus of our investigation, therefore we limit ourselves to their applications to integrability. The underlying idea is that, sometimes, the thermodynamics of a quantum model can be better controlled than its classical counterpart. Therefore, one uses quantum physics to shed new light on the classical world, going upstream to the usual common sense. Semiclassical limits of integrable models appeared quite early in the literature: curiously, these ideas have been applied to the very sine-Gordon model [37,40], but an overlooked subtlety in accounting for the entropy of states led to incorrect results, equivalent with the soliton-gas picture. To the best of our knowledge, semiclassical limits of thermodynamics have been more recently brought to the forefront in

Ref. [57], finding a fertile terrain due to the uprising interest in nonequilibrium many-body physics. Shortly after, these ideas have been extended to sudden quenches [59] and then fused with Generalized Hydrodynamics to tackle nonequilibrium protocols in several models [53, 58, 66]. Before turning to examining sine-Gordon with this rich toolbox, we recollect some basic notions of the quantum model.

**The quantum sine-Gordon** — Replacing classical fields with operators in the sine-Gordon Hamiltonian has far-fetching consequences. Here, we only focus on the main ingredients needed for our purposes. The interested reader can refer to Ref. [55] and references therein. As mentioned before, the quantum Hamiltonian is the same as the classical one (1), provided the fields are promoted to operators. For later convenience, we redefine the quantum interaction as $g \to g_q$. The quantum model is best discussed in terms of the coupling $\xi = \frac{cg^2}{8\pi}\left(1 - \frac{cg^2}{8\pi}\right)^{-1}$, which determines the excitations' spectrum of the theory. For $\xi > 1$ the only excitations are kinks and anti-kinks: their mass is heavily renormalized by quantum effects with respect to the classical result (2) $M \to M_q$ and has been computed in Ref. [67]. For smaller couplings $\xi$, breathers appear in the spectrum with quantized masses

$$m_{q;n} = 2M_q \sin(\frac{\pi}{2}n\xi), \qquad n = 1, 2, ..., N \qquad N = \lceil \xi^{-1} \rceil. \tag{9}$$

The other main ingredient we need is the scattering matrix, which has been exactly computed in Ref. [23]. When two breathers meet, their scattering is purely transmissive, but they accumulate a non-trivial phase due to interactions as

$$S_{n,m}(\theta) = \frac{\sinh\theta + i\sin((n+m)\pi\xi/2)}{\sinh\theta - i\sin((n+m)\pi\xi/2)}\frac{\sinh\theta + i\sin(|n-m|\pi\xi/2)}{\sinh\theta - i\sin(|n-m|\pi\xi/2)} \times$$
$$\prod_{k=1}^{\min(n,m)-1} \frac{\sin^2\left((|n-m|+2k)\pi\xi/4 - i\theta/2\right)}{\sin^2\left((|n-m|+2k)\pi\xi/4 + i\theta/2\right)}\frac{\cos^2\left((m+n-2k)\pi\xi/4 + i\theta/2\right)}{\cos^2\left((m+n-2k)\pi\xi/4 - i\theta/2\right)}. \tag{10}$$

Similarly, kinks are transmitted through breathers with a phase $S_n(\theta)$

$$S_n(\theta) = \frac{\sinh\theta + i\cos(n\pi\xi/2)}{\sinh\theta - i\cos(n\pi\xi/2)}\prod_{k=1}^{n-1}\frac{\sin^2\left((n-2k)\pi\xi/4 - \pi/4 + i\theta/2\right)}{\sin^2\left((n-2k)\pi\xi/4 - \pi/4 - i\theta/2\right)}. \tag{11}$$

When it comes to kinks, the scattering processes become more complicated. While kink-kink and antikink-antikink scattering is transmissive with scattering matrix $S(\theta) = -\exp\left[-i\int_0^\infty \frac{dt}{t}\frac{\sinh(\pi t(1-\xi)/2)}{\sinh(\pi\xi t/2)\cosh(\pi t/2)}\sin(\theta t)\right]$, the scattering of kinks with antikinks has a more quantum mechanical flavor and they can be either transmitted (as in the classical case) or reflected, with amplitudes that are respectively

$$S_T(\theta) = \frac{\sinh(\xi^{-1}\theta)}{\sinh((i\pi - \theta)\xi^{-1})}S(\theta), \qquad S_R(\theta) = i\frac{\sin(\pi\xi^{-1})}{\sinh((i\pi - \theta)\xi^{-1})}S(\theta). \tag{12}$$

Armed with the knowledge of the quantum model, we will now turn to the semiclassical limit.

## 3.1 Taking the semiclassical limit

The semiclassical limit is attained in the regime of high occupation number and weak inter-actions. The proper scaling can be pinned down just by looking at the partition function (or the propagator) in a path integral formalism, without even referring to integrability. We will not repeat these passages since they have already been extensively discussed in the literature, see e.g. Refs. [57, 59], and simply quote the sought scaling. For the sake of convenience, we take the limit by introducing a fictitious Planck constant $\hbar$ which will be later sent to zero. The limit is then obtained by the simultaneous rescaling

$$g_q = \sqrt{\hbar} g \,, \qquad \langle Q_q \rangle = \frac{1}{\hbar} \langle Q \rangle \,. \tag{13}$$

Above, $Q_q$ and $Q$ are meant to be any conserved charge of the quantum and classical model, respectively. While the scaling of conserved charges is crucial in analyzing thermodynamic properties, the scaling of the interactions is already enough to tackle the few-body sector.

**The scaling of the spectrum and scattering data** — As a propaedeutical analysis, we begin by analyzing the particle spectrum and scattering properties. The fist oddity the quantum model exhibits in comparison with the classical one, is the fact that kink-antikink can both be transmitted or reflected. However, in the semiclassical limit $\xi \propto \hbar \to 0$, by taking the ratio between the reflected and transmitted amplitudes, it is immediate to realize that only transmission is possible. Indeed, in this limit $\lim_{\hbar \to 0} S_R(\theta)/S_T(\theta) = 0$. Therefore, quantum (anti)kinks behave as their classical counterparts, provided we show that the quantum scattering shift matches the classical one. We postpone this point after we have analyzed the breather spectrum. The mapping from the quantum breather index and the classical spectral parameter is

$$\sigma \leftrightarrow n\xi \simeq \frac{n\hbar}{s_{\max}} \,, \qquad s_{\max} \equiv \frac{8\pi}{cg^2} \,, \tag{14}$$

where we kept only the leading behavior in $\hbar$. Above, we defined the parameter $s_{\max}$ that has the interpretation of being the maximum number of breathers (divided by $\hbar^{-1}$) sine-Gordon would have upon requantization. With this identification the quantum mass law (9) starts resembling the classical one (6) provided the soliton mass scales correctly. At small couplings, the exact quantum kink mass [67] has the same form of the classical result (2), but with the quantum interaction $g_q$ in place of the classical one $g$. Therefore, by plugging $\hbar$ we establish the identification $M_q = \hbar^{-1} M$, which results in the same scaling for the breather masses. The $\hbar-$divergence of the overall mass scale is not a fluke of our mapping, but it will be crucial to achieve the correct scaling of the conserved quantities (13). When two quantum wave packets scatter, they experience a Wigner phase shift dictated by the scattering phase [45, 68]: the quantum scattering shift is readily extracted by the logarithmic derivative of the scattering matrix. More precisely, for the breathers' scattering one has $\varphi_{q;n,n'}(\theta) = i\partial_\theta \log S_{n,n'}(\theta)$: in the semiclassical limit, we must recover the classical scattering shift. This can be easily done from Eq. (10), after having taken the logarithm and the rapidity derivative, by replacing the sum with an integral according to Eq.(14). We leave the details of the calculation to Appendix A, where we find

$$\varphi_{\sigma,\sigma'}(\theta) \simeq \hbar\varphi_{q;n,n'} \,, \quad \varphi_\sigma(\theta) \simeq \hbar\varphi_{q;n} \,, \quad \varphi(\theta) \simeq \hbar\varphi_q \,. \tag{15}$$

Here, $\sigma(\sigma')$ is linked to $n(n')$ through Eq. (14). With these building blocks, we now move to the thermodynamic limit.

**The scaling of the excitations' densities and phase-space** — When an extensive number of excitations is present in the system, one describes macrostates in terms of a quasiparticle distribution. Therefore, one introduces the so-called root densities $\rho_K(\theta)$ and $\rho_{\bar{K}}(\theta)$ describing the density of kinks and antikinks respectively, and $\rho_\sigma(\theta)$ to account for the breathers. Similar quantities are introduced in the quantum world within the framework of Thermodynamic Bethe Ansatz [29] and describe both equilibrium states, and more general non-equilibrium steady states in the form of Generalized Gibbs Ensambles [20]. We now use the scaling of the charges (13) to infer the correspondence of root densities. Due to locality, conserved charges act additively on quasiparticles leading to the expression

$$L^{-1}\langle Q_q\rangle = \int \mathrm{d}\theta\,\{q_{q;K}(\theta)\rho_{q;K}(\theta) + q_{q;\bar{K}}(\theta)\rho_{q;\bar{K}}(\theta)\} + \sum_n \int \mathrm{d}\theta\, q_{q;n}(\theta)\rho_{q;n}(\theta)\,. \tag{16}$$

Above, $q(\theta)$ is called the charge eigenvalue and, similarly to the scattering shift and the energy, the charge eigenvalues of the breathers obtained by analytic continuation $q_{q;n}(\theta) = q_{q;K}\left(\theta + i\frac{\pi}{4}(1-n\xi)\right) + q_{q;\bar{K}}\left(\theta - i\frac{\pi}{4}(1-n\xi)\right)$. In particular, sine-Gordon is known to have conserved charges for all odd values of spin $s$ of the form $q_{q;K}(\theta) = q_{q;\bar{K}}(\theta) = M_q c^2 e^{s\theta}$ [55]: by setting $s = \pm 1$, for example, one recovers linear combinations of energy and momentum.

The divergence of the overall quantum mass scale $M_q \propto \hbar^{-1}$ is crucial in the proper limit: Eq. (16) converges to the classical expression if we set

$$\rho_K(\theta) = \lim_{\hbar \to 0} \rho_{q;K}(\theta)\,, \qquad \rho_\sigma(\theta) = \lim_{\hbar \to 0} \hbar^{-1}\rho_{q;n}(\theta)\Big|_{\sigma = n\hbar/s_{\max}}\,. \tag{17}$$

Therefore, from Eq. (16) we obtain

$$L^{-1}\langle Q\rangle = \int \mathrm{d}\theta\,\{q_K(\theta)\rho_K(\theta) + q_{\bar{K}}(\theta)\rho_{\bar{K}}(\theta)\} + \int_0^1 \mathrm{d}\sigma\, s_{\max} \int \mathrm{d}\theta\, q_\sigma(\theta)\rho_\sigma(\theta)\,, \tag{18}$$

where for the odd-spin classical charges we have $q_K(\theta) = q_{\bar{K}}(\theta) = Mc^2 e^{s\theta}$ and $q_\sigma(\theta) = q_K(\theta + i\frac{\pi}{4}(1-\sigma)) + q_{\bar{K}}(\theta - i\frac{\pi}{4}(1-\sigma))$. In principle, the integration measure $s_{\max}$ could be absorbed in a redefinition of $\rho_\sigma$, but we prefer to keep it explicit. Besides conservation laws, another key ingredient in describing thermodynamics is the total phase-space density $\rho^t$: in the quantum regime, this is defined through a set of integral equations [29]. We obtain a finite scaling from the quantum to the classical case by setting

$$\rho_K^t(\theta) = \lim_{\hbar \to 0} \hbar \rho_{q;K}^t(\theta)\,, \qquad \rho_\sigma^t(\theta) = \lim_{\hbar \to 0} \hbar \rho_{q;n}^t(\theta)\Big|_{\sigma = n\hbar/s_{\max}}\,. \tag{19}$$

The resulting equations in the classical field theory are thus

$$\rho_K^t(\theta) = \frac{cM}{2\pi}\cosh\theta - \int \frac{\mathrm{d}\theta'}{2\pi}\varphi(\theta-\theta')(\rho_K(\theta')+\rho_{\bar{K}}(\theta')) - \int_0^1 \mathrm{d}\sigma\, s_{\max} \int \frac{\mathrm{d}\theta'}{2\pi}\varphi_\sigma(\theta-\theta')\rho_\sigma(\theta') \tag{20}$$

$$\rho_\sigma^t(\theta) = \frac{cm_\sigma}{2\pi}\cosh\theta - \int \frac{\mathrm{d}\theta'}{2\pi}\varphi_\sigma(\theta-\theta')(\rho_K(\theta')+\rho_{\bar{K}}(\theta')) - \int_0^1 \mathrm{d}\sigma'\, s_{\max} \int \frac{\mathrm{d}\theta'}{2\pi}\varphi_{\sigma,\sigma'}(\theta-\theta')\rho_{\sigma'}(\theta')\,. \tag{21}$$

Above, we omit the equation for the antikinks since it is the same as the kinks' one. In passing, we notice the physical interpretation of these equations [68]: $\rho^t$ is nothing else than

the reduced phase-space of a gas of extended particles with rapidity-dependent length, the latter being set by the scattering kernel. Indeed, the very same expression is postulated by the soliton-gas picture [34, 39].

The root density and the total root densities are the macroscopic variables upon which thermodynamics is built. However, they come as two independent equations and do not yet describe, for example, thermal states. To do this, one should bind the two through a minimization of a proper free energy: this is where our procedure and the soliton-gas picture begin to differ.

**The entropy and large-soliton regularization** — Let us consider the very concrete problem of determining the root density of a thermal state with some inverse temperature $\beta$. By starting with the quantum sine-Gordon, one defines a free energy

$$A_q = \beta_q \langle H_q \rangle - \mathcal{S}_q[\{\rho_{q;K}, \rho_{q;\bar{K}}, \rho_{q;n}\}], \tag{22}$$

where the entropy $\mathcal{S}_q$ is nothing else than the so called Yang-Yang entropy [29]

$$\mathcal{S}_q[\{\rho_{q;K}, \rho_{q;\bar{K}}, \rho_{q;n}\}] = L \int d\theta \left\{ \rho_{q;K}^t s\left(\frac{\rho_{q;K}}{\rho_{q;K}^t}\right) + \rho_{q;\bar{K}}^t s\left(\frac{\rho_{q;\bar{K}}}{\rho_{q;\bar{K}}^t}\right) + \sum_n \rho_{q;n}^t s\left(\frac{\rho_{q;n}}{\rho_{q;n}^t}\right) \right\}, \tag{23}$$

where for notation convenience we kept implicit the rapidity dependence of the roots and defined the function $s(x) = -x \log x - (1-x) \log(1-x)$. The classical free energy is readily obtained from the quantum expression (22) by using our semiclassical scaling. Since the expectation value of the energy diverges $\sim \hbar^{-1}$, the quantum temperature should be likewise rescaled to attain a finite value. However, our main focus is now the entropy

$$\mathcal{S}_q[\{\rho_{q;K}, \rho_{q;\bar{K}}, \rho_{q;n}\}] \stackrel{\hbar \to 0}{=} L \int d\theta \left\{ \rho_K \left[ 1 - \log\left(\frac{\rho_K}{\rho_K^t}\right) \right] + \rho_{\bar{K}} \left[ 1 - \log\left(\frac{\rho_{\bar{K}}}{\rho_{\bar{K}}^t}\right) \right] + \right.$$

$$\left. + \int_{\delta_\hbar}^1 d\sigma \, s_{\max} \rho_\sigma \left[ 1 - \log\left(\frac{\rho_\sigma}{\rho_\sigma^t}\right) \right] - \log h \left[ \rho_K + \rho_{\bar{K}} + 2 \int_{\delta_\hbar}^1 d\sigma \, s_{\max} \rho_\sigma \right] \right\}. \tag{24}$$

Above, we can recognize several terms: For example, kinks contribute to the entropy with $\rho_{\bar{K}} \left[ 1 - \log\left(\rho_{\bar{K}}/\rho_{\bar{K}}^t\right) \right]$, which is nothing else than the entropy of classical particles (i.e. with Maxwell-Boltzmann statistics) in a renormalized volume set by the total root density. Similar contributions are associated to antikinks and breathers. Nonetheless, a further $\propto \log \hbar$ term seems to prevent a straightforward semiclassical limit. However, it turns out to be crucial in order to get a well-defined expression in the classical model, as we now discuss. Above, we introduced a $\hbar$-dependent cutoff in the breathers' spectral parameter $\sigma > \delta_\hbar$. Naively, one could have imposed $\delta_\hbar = 0$, but when converting the sum over the quantum breathers to an integral, it should not be forgotten that in the quantum-classical correspondence (14) $\sigma$ has a lower bound $\propto \hbar$. As $\sigma$ approaches zero, the spatial extend of the breather grows, and $\delta_\hbar$ is exactly the ingredient we need to regularize the large-soliton problem faced by the soliton-gas picture. Indeed, in the soliton-gas picture [34, 39] and in previous semiclassical limits of sine-Gordon [36, 37, 40], the $\log \hbar$ term in Eq. (24) has been ovelooked and the cutoff $\delta_\hbar$ was absent. We notice that the $\propto \log \hbar$ term can be seen as the chemical potential introduced in Ref. [38]: however, in our case it is a divergent quantity, while in the quoted reference it is eventually sent to zero.

In contrast, one should fix $\delta_\hbar$ with the following strategy: for finite $\hbar$, minimize the free energy by finding the saddle point $\delta A_q/\delta\rho = 0$, then impose that the resulting equations remain consistent (i.e. non-singular) as $\hbar \to 0$. The dangerous terms are those where the breathers' spectral parameter is small. In spirit, this computation closely follows the steps of a previous semiclassical limit of the attractive Non-Linear-Schrödinger equation [53]. Therefore, we only report the result, leaving an overview of the computation to Appendix B. In particular, we find that $\delta_\hbar$ should be fixed by asking

$$\log\left(\hbar^{-1}\delta_\hbar s_{\max}\right) = 1 \tag{25}$$

leading to the following equations describing the classical Thermodynamic Bethe Ansatz (we report only those for kinks and breathers, the one for the antikinks is the same as the kinks' one)

$$\sigma\varepsilon_\sigma(\theta) = -2 + \beta c^2 \frac{m_\sigma}{\sigma}\cosh\theta + \frac{1}{\sigma}\int\frac{\mathrm{d}\theta'}{2\pi}\varphi_\sigma(\theta-\theta')(e^{-\varepsilon_K}+e^{-\varepsilon_{\bar{K}}})+$$
$$+\frac{1}{\sigma}\int\frac{\mathrm{d}\theta'}{2\pi}\int_0^1\mathrm{d}\sigma'\,\varphi_{\sigma,\sigma'}(\theta-\theta')\frac{e^{-(\sigma')^2\varepsilon_{\sigma'}(\theta')}-1}{s_{\max}(\sigma')^2}\,, \tag{26}$$

$$\varepsilon_K(\theta) = \log s_{\max} - 1 + \beta M c^2\cosh\theta + \int\frac{\mathrm{d}\theta'}{2\pi}\varphi(\theta-\theta')(e^{-\varepsilon_K}+e^{-\varepsilon_{\bar{K}}})+$$
$$+\int\frac{\mathrm{d}\theta'}{2\pi}\int_0^1\mathrm{d}\sigma\varphi_\sigma(\theta-\theta')\frac{e^{-\sigma^2\varepsilon_\sigma(\theta')}-1}{s_{\max}\sigma^2}\,, \tag{27}$$

where we introduced the effective energies $\varepsilon$ and, for later use, the filling functions $\vartheta$ as

$$\vartheta_K(\theta) = e^{-\varepsilon_K(\theta)} = \frac{\rho_K(\theta)}{\rho_K^t(\theta)}\,,\qquad \vartheta_\sigma(\theta) = e^{-\sigma^2\varepsilon_\sigma(\theta)} = (s_{\max}\sigma)^2\frac{\rho_\sigma(\theta)}{\rho_\sigma^t(\theta)}\,, \tag{28}$$

and an analogous definition holds for the antikinks. Before moving further, a few comments are due: commonly in the literature, the filling functions are defined as the ratio between the root density and the total root density, but this choice is not convenient in our case. Indeed, $\frac{\rho_\sigma(\theta)}{\rho_\sigma^t(\theta)} \simeq 1/(\sigma s_{\max})^2$ as $\sigma \to 0$ (see Appendix B). Hence, we define the non-singular part as the filling function. With our definition, $\lim_{\sigma\to 0}\vartheta_\sigma(\theta) = 1$. This is also reflected in our definition of the effective energy $\varepsilon$, since $\lim_{\sigma\to 0}\varepsilon_\sigma(\theta) < +\infty$. Secondarily, even if we insist in using the more conventional notation, the equations (26-27) differ from those derived within the soliton-gas picture by *i)* an extra contribution to the source term and *ii)* the presence of a "$-1$" term in the integrals over breathers. Notice that this modification guarantees that $(e^{-\sigma^2\varepsilon_\sigma(\theta)}-1)/(s_{\max}\sigma^2)$ remains finite for $\sigma \to 0$.

## 3.2 The expectation value of the vertex operator

After having derived the equations governing the thermodynamics, we would like to test their prediction on observables that can be analytically computed in some limiting cases and numerically tabulated in all regimes. Conserved charges seem to be the ideal candidates, given their simple expression in terms of the root densities (18). For example, one could focus on the energy. However, on thermal states the expectation value of the Hamiltonian diverges,

due to the UV-black body catastrophe (see also Refs. [53,58]). Therefore, we rather revert to other observables such as the expectation value of the vertex operator $\langle\cos(g\phi)\rangle$. Within the quantum case, this observable can be computed in a closed form for any Generalized Gibbs Ensemble, and thus on thermal states as well, by observing that $\cos(g\phi)$ is proportional to the derivative of the Hamiltonian w.r.t. the bare mass $m$, and then using the Hellmann-Feynman theorem [69]. Finally, the semiclassical limit is readily taken: the details are discussed in Appendix C.

Before reporting the formula, we need to preliminary define the so-called dressing operation in the classical theory. Due to the singular behavior of the canonical definition of the filling function discussed previously, it is convenient to redefine the standard expression of the dressing operation by removing the most singular part. Let us consider a triplet of test functions $\{\tau_K(\theta), \tau_{\bar{K}}(\theta), \tau_\sigma(\theta)\}$. One defines then the dressing operation $\{\tau_K(\theta), \tau_{\bar{K}}(\theta), \tau_\sigma(\theta)\} \to \{\tau_K^{\mathbf{dr}}(\theta), \tau_{\bar{K}}^{\mathbf{dr}}(\theta), \tau_\sigma^{\mathbf{dr}}(\theta)\}$ as the solution of the following linear integral equations

$$\sigma\tau_\sigma^{\mathbf{dr}}(\theta) = \frac{\tau_\sigma(\theta)}{\sigma} - \frac{1}{\sigma}\int\frac{\mathrm{d}\theta'}{2\pi}\varphi_\sigma(\theta-\theta')[\vartheta_K(\theta')\tau_K^{\mathbf{dr}}(\theta') + \vartheta_{\bar{K}}(\theta')\tau_{\bar{K}}^{\mathbf{dr}}(\theta')]$$
$$- \frac{1}{\sigma}\int_0^1\frac{\mathrm{d}\sigma'}{s_{\max}}\int\frac{d\theta'}{2\pi}\varphi_{\sigma,\sigma'}(\theta-\theta')\vartheta_{\sigma'}(\theta')\tau_{\sigma'}^{\mathbf{dr}}(\theta'),\quad(29)$$

$$\tau_K^{\mathbf{dr}}(\theta) = \tau_K(\theta) - \int\frac{\mathrm{d}\theta'}{2\pi}\varphi(\theta-\theta')[\vartheta_K(\theta')\tau_K^{\mathbf{dr}}(\theta') + \vartheta_{\bar{K}}(\theta')\tau_{\bar{K}}^{\mathbf{dr}}(\theta')]+$$
$$- \int_0^1\frac{\mathrm{d}\sigma}{s_{\max}}\int\frac{d\theta'}{2\pi}\varphi_\sigma(\theta-\theta')\vartheta(\theta',\sigma)\tau_\sigma^{\mathbf{dr}}(\theta').\quad(30)$$

These classical dressing equations naturally emerge as the semiclassical limit of the quantum ones (see Appendix C). Notice the connection between the total root densities and the dressed derivative of the momenta $2\pi\rho_K^t(\theta) = (\partial_\theta p_K)^{\mathbf{dr}}(\theta)$ and $2\pi\rho_\sigma^t(\theta) = \sigma^2(\partial_\theta p_\sigma)^{\mathbf{dr}}(\theta)$, with $p_K(\theta) = Mc\sinh\theta$ and $p_\sigma(\theta) = m_\sigma c\sinh\theta$ the kink's and breather's momenta, respectively. The expectation value of the vertex operator can be then computed with the following expression (we silenced the obvious $\theta-$dependence of the functions for the sake of notation)

$$2\frac{m^2c^2}{g^2}\langle 1-\cos(g\phi)\rangle = \int\frac{\mathrm{d}\theta}{2\pi}\int_0^1\frac{\mathrm{d}\sigma}{s_{\max}}\,m_\sigma c(\cosh\theta\epsilon_\sigma^{\mathbf{dr}} - c\sinh\theta p_\sigma^{\mathbf{dr}})\vartheta_\sigma+$$
$$\int\frac{\mathrm{d}\theta}{2\pi}Mc(\cosh\theta\epsilon_K^{\mathbf{dr}} - c\sinh\theta p_K^{\mathbf{dr}})\vartheta_K + \int\frac{\mathrm{d}\theta}{2\pi}Mc(\cosh\theta\epsilon_{\bar{K}}^{\mathbf{dr}} - c\sinh\theta p_{\bar{K}}^{\mathbf{dr}})\vartheta_{\bar{K}}.\quad(31)$$

We are finally in the right position to test our thermodynamic prediction: in Figure 2 we numerically solve the set of integral equations defining thermal states (26,27), then plug the result in Eq. (31). The so-obtained data are compared against an ab-initio numerical evaluation of the vertex operator on thermal states obtained through the Transfer Matrix approach [70,71], finding perfect agreement for all temperature regimes. Despite the similar name, this transfer matrix method is different from the one we previously mentioned in the context of integrability and we shortly discuss it in Appendix D. We stress that we perfectly capture the low-temperature regime, where other methods gave incorrect results, see Ref. [40] for a compact overview. It turns out that the low temperature regime is even amenable of an analytical analysis: this is a very instructive calculation that *i)* shows the importance of

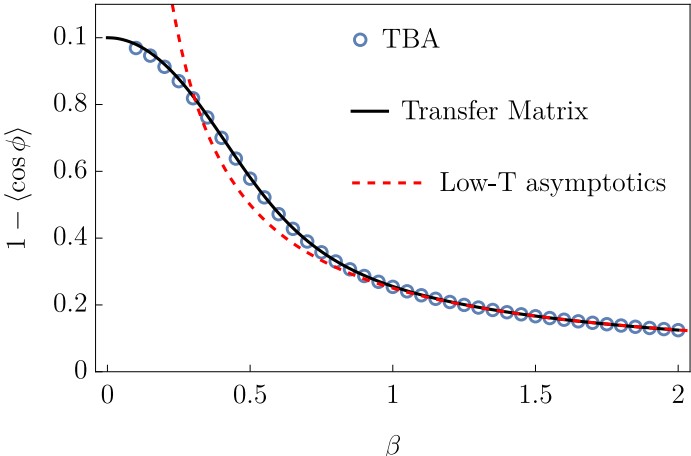

Figure 2: **The vertex operator at equilibrium.** We compare the temperature dependence of the expectation value of the vertex operator as obtained by solving the Thermodynamic Bethe Ansatz (TBA) equations Eq. (26-27) [symbols] against the numeric predictions of the Transfer-Matrix approach [full black line]. As a comparison, we plot the low temperature asymptotics dragged from the Klein-Gordon field theory [red dashed line]. While at large $\beta$ the three curves collapse, at higher temperature a clear deviation of the Klein-Gordon approximation is observed, while the agreement between the Transfer Matrix and the exact prediction remains excellent. At equilibrium, the mass scale $m$, sound velocity $c$ and interactions $g$ can be absorbed in a rescaling of distances and fields. Therefore, we set $m = c = g = 1$ in the above leaving $\beta$ as the only free parameter. Numerical methods are discussed in Appendix D.

cutoff $\delta_\hbar$ in the entropy, leading us to the correct equations (26,27), and *ii)* helps shining light on the role of the, so far missing, radiative modes. Indeed, at low temperatures the phase field is expected to be pinned down at one of the potential minima, let's say $\phi = 0$. Hence, upon Taylor expanding the cosine potential, the sine-Gordon Hamiltonian (1) is well-approximated by the Klein-Gordon model $H_{KL} = \int \mathrm{d}x\, \{\frac{1}{2}c^2\Pi^2(x) + \frac{1}{2}(\partial_x\phi)^2 + \frac{m^2c^2}{2}\phi^2\}$. The Klein-Gordon field theory has only a single radiative mode, thus is distributed according to a Rayleigh-Jeans distribution $1/[\beta \times (\text{energy})]$, leading to

$$2\frac{m^2c^2}{g^2}\langle 1 - \cos(g\phi)\rangle \overset{\beta\to\infty}{\simeq} m^2c^2\langle\phi^2\rangle_{KL} = m^2c^3\int\frac{\mathrm{d}\theta}{2\pi}\frac{1}{\beta mc^2\cosh\theta}\,. \tag{32}$$

We will now recover this result from sine-Gordon. First, we simplify Eqs. (26,27): for small temperatures, the kink-antikink fillings are exponentially suppressed, hence we can entirely neglect them. Likewise, only breathers with small $\sigma$ will contribute: therefore, we can safely approximate $m_\sigma \sim M\pi\sigma$. We furthermore can make additional approximations: we toss away the "$-2$" term in Eq. (26) which is subleading with respect to $\beta Mc^2\pi$. Since only breathers with small spectral parameter are important, we can approximate $\varphi_{\sigma,\sigma'}(\theta)$ with the proper limit. In particular, the scattering shift collapses to a Dirac$-\delta$ in the rapidity space $\varphi_{\sigma,\sigma'}(\theta - \theta') \overset{\sigma,\sigma'\to 0}{=} 4\pi s_{\max}\min(\sigma,\sigma')\delta(\theta - \theta')$. Lastly, by neglecting further terms subleading

w.r.t. $\beta Mc^2\pi$, we can extend the integration domain of the spectral parameter to the whole real axis. We finally reach the simplified equation

$$\sigma\varepsilon_\sigma(\theta) \stackrel{\beta\to+\infty}{=} \beta c^2\pi M\cosh\theta + \int_0^\infty \mathrm{d}\sigma'\, \frac{2\min(\sigma,\sigma')}{\sigma} \frac{e^{-(\sigma')^2\varepsilon_{\sigma'}(\theta)}-1}{(\sigma')^2}\,. \tag{33}$$

Albeit non-trivial, these equations can be exactly solved [35, 53], leading to an exact low-temperature analytical expression for the filling function (28)

$$\vartheta_\sigma(\theta) \stackrel{\beta\to+\infty}{=} \frac{(\beta\sigma Mc^2\pi\cosh\theta)^2}{4\sinh^2\left(\frac{\beta\sigma}{2}Mc^2\pi\cosh\theta\right)}\,. \tag{34}$$

We now revert to the dressing equations (29) by invoking the same approximations and with some simple manipulations, the dressing equations supplemented with Eq. (34) are readily recast in a scaling form

$$\tau_\sigma^{\mathbf{dr}}(\theta) \stackrel{\beta\to+\infty}{=} \left[\lim_{\sigma\to0}\frac{\tau_\sigma(\theta)}{\sigma}\right]\frac{u\left(\beta\sigma Mc^2\pi\cosh\theta\right)}{\sigma^2\left(\beta Mc^2\pi\cosh\theta\right)}\,, \tag{35}$$

where $u(x)$ satisfies

$$u(x) = x - \int \mathrm{d}y\, \frac{2\min(x,y)}{4\sinh^2(y/2)}u(y)\,. \tag{36}$$

An explicit solution can be found by taking a second derivative on both sides and obtaining the differential equation $x\sinh^2(x/2)\frac{\mathrm{d}^2u(x)}{\mathrm{d}^2x} = u(x)$, which leads to $u(x) = x\coth(x/2) - 2$.

We use this solution in Eq. (31) and, upon neglecting kinks and antinkins, we find

$$2\frac{m^2c^2}{g^2}\langle(1-\cos(g\phi))\rangle = m^2c^3\int\frac{\mathrm{d}\theta}{2\pi}\frac{1}{\beta mc^2\cosh\theta}\int_0^\infty \mathrm{d}x\frac{x^2\coth(x/2)-2x}{4\sinh^2\left(\frac{x}{2}\right)}\,. \tag{37}$$

Noticing the identity $\int_0^\infty \mathrm{d}x\frac{x^2\coth(x/2)-2x}{4\sinh^2\left(\frac{x}{2}\right)} = 1$, we finally have the the equality of the above expression with Eq. (32). We have therefore shown the consistency of the low-temperature regime of our sine-Gordon thermodynamics. Notice that the "$-1$" in the integrand of Eq. (33), which has been overlooked in the soliton-gas picture, is crucial to obtain the correct result. Before finally moving to the nonequilibrium scenario, we would like to further comment on the radiation-soliton interplay: our analysis of the vertex operator shows how the breathers for small spectral parameter (each of them having solitonic nature) can collectively behave as the radiative mode of the Klein-Gordon field theory. However, since all the modes contribute to the vertex operator, how this reorganization happens is not evident. A better understanding can be achieved by computing, in the low-temperature limit, the total energy carried by the modes with rapidity $\theta$ summing over the spectral parameter

$$\int_0^1 \mathrm{d}\sigma s_{\max} m_\sigma c^2\cosh\theta\rho_\sigma(\sigma) \stackrel{\beta\to\infty}{=} \frac{mc\cosh\theta}{2\pi\beta}\,. \tag{38}$$

We can compare this result with the thermal mode occupation of Klein-Gordon: the mode density at fixed rapidity $n(\theta)$ is populated as $n(\theta) = \frac{\mathrm{d}[\frac{1}{2\pi}mc\sinh(\theta)]}{\mathrm{d}\theta}\frac{1}{\beta mc^2\cosh\theta}$, where the $1/[\beta\times$ (energy)] term comes from the Rayleigh-Jeans distribution of radiation in the momentum space, while the prefactor is the Jacobian to pass from momenta to rapidities. Multiplying the Klein-Gordon mode density by the energy $mc^2\cosh\theta$ we match Eq. (38), undoubtedly showing that radiative modes can be thought of as a "condensation" of extended solitons with the same rapidity.

# 4 From thermodynamics to transport: Generalized Hydrodynamics

We built the thermodynamics of the classical sine-Gordon theory, but the same concepts can be extended to nonequilibrium states, such as homogeneous quantum quenches [72, 73], and transport settings [46] through Generalized Hydrodynamics. Here, we take this second path and study the paradigmatic transport setting, namely the partitioning protocol [54]. To this end, we need to write down the proper hydrodynamic equations. As it is assumed within the nowadays-standard method of Generalized Hydrodynamics [22, 32, 33], within a local density approximation the root density is promoted to be a weakly space-time dependent function that locally parametrizes the state. In the simplest scenario where the sine-Gordon Hamiltonian is kept homogeneous in space and constant in time, while the only inhomogeneity is carried by the initial state, the large scale dynamics is captured by the following continuity equations

$$\partial_t \rho_\sigma(\theta, t, x) + \partial_x[v_\sigma^{\text{eff}} \rho_\sigma(\theta, t, x)] = 0 \,, \qquad \partial_t \rho_K(\theta, t, x) + \partial_x[v_K^{\text{eff}} \rho_K(\theta, t, x)] = 0 \,. \qquad (39)$$

As usual, we omit the antikinks' equation since it is analogous to the kinks' one. Above, the effective velocity $v^{\text{eff}}$ is a state-dependent renormalized velocity that is defined by properly "dressing" the group velocity, thus obtaining $v_\sigma^{\text{eff}} = (\partial_\theta \epsilon_\sigma)^{\text{dr}}/(\partial_\theta p_\sigma)^{\text{dr}}$ and $v_K^{\text{eff}} = (\partial_\theta \epsilon_K)^{\text{dr}}/(\partial_\theta p_K)^{\text{dr}}$ (although exceptions to this definition are known [74]). The dressing is performed by using the root-density at a given space and time position, such that the effective velocity gets an implicit dependence in space and time as well. The equations (39) has been proposed for the first time in quantum models in the seminal papers Refs. [32, 33], but further progresses have been made including corrections beyond the Eulerean scale [75–78] and inhomogeneities and time-dependence in the Hamiltonian [47, 48]. We leave these questions for future developments: the interested reader can find these extensions, together with many other results and applications, in the recent review paper Ref. [22]. Eq. (39) can also be equivalently recast in more convenient equations for the filling function $\vartheta$, which is also promoted to be a weakly space-time dependent function

$$\partial_t \vartheta_\sigma(\theta, t, x) + v_\sigma^{\text{eff}} \partial_x[\vartheta_\sigma(\theta, t, x)] = 0 \,, \qquad \partial_t \vartheta_K(\theta, t, x) + v_K^{\text{eff}} \partial_x[\vartheta_K(\theta, t, x)] = 0 \,. \qquad (40)$$

Since $v^{\text{eff}}$ is state-dependent itself, the equivalence of the two formulations may appear not evident, but passing from one to the other requires standard manipulation already discussed in the literature [32, 33] and thus is not reported here. Of course, we checked that these hydrodynamic equations are consistent with taking the semiclassical limit of the Generalized Hydrodynamics of the quantum sine-Gordon model. As anticipated, we aim to use these equations to study the paradigmatic partitioning protocol: in this setting, the state is initialized in two different halves, each of them described by a Genealized Gibbs Ensemble and thus identified by a left(right) filling function $\vartheta^{L(R)}$. For example, thermal states at different temperatures are a common choice. Then, at $t > 0$ the two halves are joined and let to evolve with the sine-Gordon Hamiltonian, generating non-trivial currents and activating transport. While generic (non-integrable) systems usually feature diffusive behavior, integrability induces ballistic transport. This is reflected into the scale invariant solution of Eq. (40) when applied to the partitioning protocol: after a short time transient which depends on the details of the interface between the two initial halves, the evolving hydrodynamic state is not an independent function of $t$ and $x$, but only depends on the ratio $\zeta = x/(ct)$. This allows

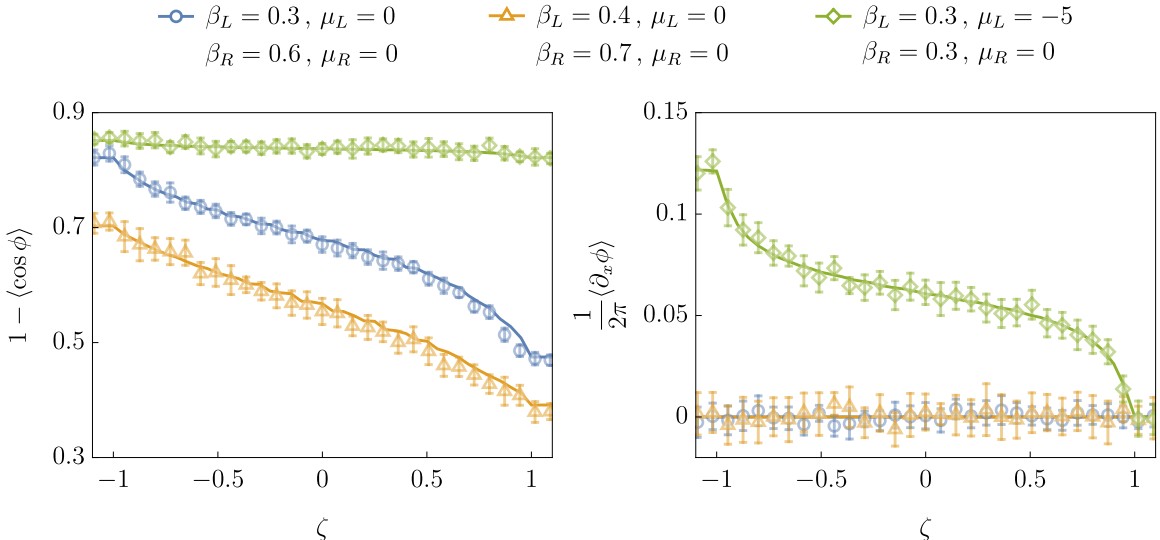

Figure 3: **Hydrodynamic transport stemming from partitioning protocols.**
We show the ray-dependent profiles $\zeta = x/(ct)$ of the vertex operator [Left] and
topological charge density [Right] in partitioning protocols. As an example, we fix
$m = c = g = 1$ and consider three cases in total: in two of them [blue dots and
yellow triangles] we choose a temperature imbalance with zero topological charge.
In the third case [green diamonds] we study equal temperatures with a chemical
potential imbalance of the topological charge. The effect of the latter barely affects
the vertex operator, but has a clear effect on the topological charge. Continuous
lines are obtained as the solution of Eq. (41), symbols are Monte Carlo data (see
D for details). Error bars are estimated as the variance of 20 independent Monte
Carlo runs with $\approx 500$ samples each. Upon close inspection, it can be seen that the
solid lines have some residual irregularities: this is a discretization error due to the
finite grid (we used $\approx 3000$ points).

one to derive an exact implicit solution to the partitioning protocol [32, 33]

$$\vartheta_\sigma(\theta, \zeta) = \Theta(v_\sigma^{\text{eff}}(\theta, \zeta) - \zeta)\vartheta_\sigma^L(\theta) + \Theta(\zeta - v_\sigma^{\text{eff}}(\theta, \zeta))\vartheta_\sigma^R(\theta) \,. \tag{41}$$

Above, $\Theta$ is the Heaviside-theta function. We report only the equations for the breathers,
but analogous equations hold for kinks and antikinks. The above solution is implicit due to
the state-dependence of the effective velocity, but an iterative numerical scheme ensures fast
convergence.

To test our hydrodynamic equations, we compare the solution of Eq. (41) against ab-initio
numerical simulations. The Transfer Matrix method used in the previous section cannot be
generalized out of equilibrium, therefore we employ Monte Carlo schemes. These techniques
are standard, but some care should be taken in properly defining the junction between the
two initial halves: we report a short overview in Appendix D. We focus on partitionings of
thermal states, possibly deformed with a non-trivial topological charge.

Indeed, in sine-Gordon a charge counting the kink-antikink difference, but insensitive to

breathers, can be defined as

$$Z = \frac{g}{2\pi} \int \mathrm{d}x \, \partial_x \phi \,. \tag{42}$$

This charge is explicitly conserved by the sine-Gordon Hamiltonian and counts how many phase-slips the system undergoes. In terms of charge eigenvalues on the excitations, which we denote with $z$, one has $z_K(\theta) = 1$, $z_{\bar{K}}(\theta) = -1$ and $z_\sigma(\theta) = 0$: thermal states can be easily deformed to take into account a non-trivial topological charge by adding a chemical potential coupled to the topological charge eigenvalue in Eqs. (26-27) $\beta H \to \beta H + \mu Z$. The topological charge gives us an extra parameter to benchmark and a further observable to test our prediction. In Figure 3 we show partitioning protocols for a variety of parameters, finding excellent agreement between hydrodynamics and Monte Carlo data, thus proving the validity of the hydrodynamic equations (40).

## 5   Conclusion and Outlook

In this paper, we revisited the thermodynamics of the classical sine-Gordon field theory, identifying a common issue in previous works due to solitons of vanishing mass, but growing spatial extension. These solitons cause subtleties in taking the thermodynamic limit, which we circumvent considering the semiclassical limit of the quantum model. In this framework, the Planck constant acts as a regulator putting a maximum cap to the size of solitons, allowing us to first safely take the thermodynamic limit and only then the classical one. Our result shows that solitons alone account for the exact thermodynamic of sine-Gordon without any explicit contribution of any radiative mode, which can be rather seen as a collective effect of light solitons. We also study transport in the form of partitioning protocols, laying the foundations for the development of the Generalized Hydrodynamics for the classical sine-Gordon model. This work opens up several interesting directions that we are eager to explore. First of all, while the physical role of the Planck constant in acting as an infrared regulator appears clear, a regularization of the large-soliton problem within the pure classical realm is highly desirable and would allow to tackle other integrable models without an obvious quantum counterpart.

Large solitons with arbitrary spatial extension are present in a plethora of classical and even quantum models with possibly far-reaching consequences. For example, this is the case in certain quantum magnets with non-abelian symmetries that feature superdiffusive transport such as the isotropic Heisenberg chain [79]: there, extended excitations with classical nature have been identified as the culprit of the anomalous transport [52]. It has been previously put forward that superdiffusion can also be understood as a fluctuating Goldstone mode of the non-abelian symmetry in an effective bath of heavier excitations [80] (see also [81]): it is natural to wonder if, in analogy to the condensation of solitons in radiative modes we observed in sine-Gordon, the Goldstone mode of the Heisenberg chain can be seen as a condensation of large solitons. Lastly, remaining within the sine-Gordon framework, we aim to use this work as a stepping stone to extend the Generalized Hyrdodynamics approach to feature inhomogeneities in the sine-Gordon's couplings: although a general hydrodynamic theory of inhomogeneous interactions has been already developed [47, 48], we expect complications arising due to binding-unbinding of solitons into breathers, similarly to the bound state recombination featured by other models [53,82,83]. This question is not of mere theoretical interest, but we expect to have ready experimental application, too, as for instance the coupled quasi-condensate experiment [12]. The latter realizes a sine-Gordon simulator with inhomogeneous

couplings and, depending on the parameters, is well described by the semiclassical regime. We envision that a full hydrodynamic treatment may shed new light on the dynamics and relaxation of the experiment.

# Data and code availability

Data analysis of Monte Carlo sampling, a Mathematica notebooks for the transfer matrix approach and a solver for the classical Thermodynamic Bethe Ansatz and partitioning are available on Zenodo upon reasonable request [84].

# Acknowledgements

We are indebited to Žiga Krajnik, Enej Ilievski, Jean-Sébastien Caux, Michael Knap and Jacopo De Nardis for useful discussions, we are particularly grateful to Žiga Krajnik for suggesting us the solution of Eq. (36). We thank Giuseppe Del Vecchio Del Vecchio, Benjamin Doyon, Marton Kormos for their insight and collaboration on related topics. In particular, we thank Žiga Krajnik and Marton Kormos for useful comments on the manuscript.

**Funding information** RK acknowledges support from the European Research Council (ERC) under ERC Advanced grant 743032 DYNAMINT. AB acknowledges support from the Deutsche Forschungsgemeinschaft (DFG, German Research Foundation) under Germany's Excellence Strategy–EXC–2111–390814868.

# A The semiclassical limit of the quantum scattering shift

Here, we explicitly show how to recover the classical breather-breather phase shift $\varphi_{\sigma,\sigma'}(\theta)$ in the semiclassical limit. As it has been explained in the main part of this paper, for the quantum scattering the definition $\varphi_{q;n,n'}(\theta) = i\partial_\theta \log S_{n,n'}(\theta)$ holds true. Therefore, the natural starting point for the semiclassical limit is the logarithm of the quantum breather-breather scattering matrix reported in Eq. (10). Focusing on the second term only (the first term will drop out in the semiclassical limit), we get

$$\log S_{n,m}(\theta) \propto \log \sum_{k=1}^{\min(n,m)-1} \frac{\sin^2\left((|n-m|+2k)\frac{\pi\xi}{4} - i\frac{\theta}{2}\right)}{\sin^2\left((|n-m|+2k)\frac{\pi\xi}{4} + i\frac{\theta}{2}\right)} \frac{\cos^2\left((m+n-2k)\frac{\pi\xi}{4} + i\frac{\theta}{2}\right)}{\cos^2\left((m+n-2k)\frac{\pi\xi}{4} - i\frac{\theta}{2}\right)} , \quad (43)$$

in which we replace the discrete parameter $n$ with the continuous spectral parameter $\sigma$ following the relation (14). This brings an explicit $\hbar$-dependence such that we can take the limit $\hbar \to 0$. The sum is then converted to an integral giving a factor $\hbar^{-1}$, which leads to the reported scaling behavior in Eq. (15) and the fact that the first factor in Eq. (10) vanishes.

Concretely, we arrive at

$$\log S_{\sigma,\sigma'}(\theta) = \int\limits_0^{\min(\sigma,\sigma')} d\tau \, s_{\max} \log \left[ \frac{\sin^2\left((-|\sigma'-\sigma|-2\tau)\frac{\pi}{4}+i\frac{\theta}{2}\right)\cos^2\left((\sigma'+\sigma-2\tau)\frac{\pi}{4}+i\frac{\theta}{2}\right)}{\sin^2\left((-|\sigma'-\sigma|-2\tau)\frac{\pi}{4}-i\frac{\theta}{2}\right)\cos^2\left((\sigma'+\sigma-2\tau)\frac{\pi}{4}-i\frac{\theta}{2}\right)} \right].$$
(44)

From here, we can use the definition of $\varphi_{\sigma,\sigma'}(\theta) = \partial_\theta i \log S_{\sigma,\sigma'}(\theta)$ and exchange the derivative $\partial_\theta$ with $\partial_\tau$ while keeping track of the appropriate factors stemming from the chain rule. Following this procedure and using trigonometric identities the integral can be carried out, and we obtain the classical phase shift

$$\varphi_{\sigma,\sigma'}(\theta) = \frac{2s_{\max}}{\pi} \log\left( \frac{[\cosh(\theta)-\cos((\sigma+\sigma')\pi/2)][\cosh(\theta)+\cos((\sigma-\sigma')\pi/2)]}{[\cosh(\theta)-\cos((\sigma-\sigma')\pi/2)][\cosh(\theta)+\cos((\sigma+\sigma')\pi/2)]} \right).$$
(45)

An analogous calculation can be done for the breather-kink phase shift $\varphi_\sigma(\theta)$. The semiclassical limit of the kink-kink phase shift $\varphi(\theta)$ boils down to

$$\varphi(\theta) = \frac{s_{\max}}{\pi} \int_0^\infty \frac{dt}{t} \frac{\sinh(t\pi/2)}{\cosh(\pi t/2)} \cos(\theta t) = \frac{s_{\max}}{\pi} \log \frac{\cosh\theta+1}{\cosh\theta-1}.$$
(46)

We did not manage to analytically perform the integral, but we numerically checked the last identity with machine precision. It is also useful to calculate the normalization of the breather-breather scattering $\int d\theta' \varphi_{\sigma,\sigma'}(\theta-\theta')$. To do so, we make use of the fact that $\varphi_{\sigma,\sigma'}(\theta)$ is the derivative of $i \log S_{\sigma,\sigma'}$, hence $\int d\theta' \varphi_{\sigma,\sigma'}(\theta-\theta') = i \log S_{\sigma,\sigma'}(\theta)\big|_{\theta=-\infty}^{\theta=+\infty}$. To evaluate this, we can conveniently use Eq. (44) and simply take the limit of logarithms of hyperbolic functions, giving

$$\int d\theta \, \varphi_{\sigma,\sigma'}(\theta'-\theta) = -2s_{\max} \int\limits_0^{\min(\sigma,\sigma')} d\tau \left( (-|\sigma'-\sigma|-2\tau)\pi - 2\pi + (\sigma'+\sigma-2\tau)\pi \right) =$$
$$= 4\pi s_{\max} \min(\sigma,\sigma').$$
(47)

Note that the extra $-2\pi$ in the integral comes from crossing a branch-cut in the complex plane.

## B  Linking the Plack constant and the large-soliton cutoff

In this Appendix, we discuss in more detail how to obtain the finite equations (26)-(27) describing the classical sine-Gordon model in thermal equilibrium. A very similar procedure was used in our previous paper to obtain the out-of-equilibrium phase of the Non-Linear Schrödinger equation with attractive interactions [53].

The starting point is the minimization of the classical version of the free energy (22). However, the semiclassical entropy (24) still has a term $\propto \log(\hbar)$ diverging in the limit $\hbar \to 0$. At the same time, the ratio $\rho_\sigma(\theta)/\rho_\sigma^t(\theta)$ develops a singularity $\propto \sigma^{-2}$, which is why we keep an explicit cutoff $\sigma > \delta_\hbar > 0$ in the first place. Inspired by our previous work on the Non-Linear Schrödinger equation [53], we define the effective energies $\varepsilon$ according to Eq. (28). By first

computing the saddle-point $\delta A_q/\delta\rho = 0$, and then expressing the so-obtained equations in terms of the effective energies, we get

$$
-\varepsilon_K(\theta) + \beta M c^2 \cosh\theta + \log\hbar + \int \frac{d\theta'}{2\pi} \varphi(\theta - \theta')(e^{-\varepsilon_K(\theta')} + e^{-\varepsilon_{\bar{K}}(\theta')})
$$
$$
+ \int \frac{d\theta'}{2\pi} \int_{\delta_\hbar}^1 d\sigma\, s_{\max}\varphi_\sigma(\theta - \theta') \frac{e^{-\sigma^2\varepsilon_\sigma(\theta')}}{(s_{\max}\sigma)^2} = 0\,, \tag{48}
$$

$$
-\sigma^2\varepsilon_\sigma(\theta) - 2\log(s_{\max}\sigma) + \beta c^2 m_\sigma \cosh\theta + 2\log\hbar + \int \frac{d\theta'}{2\pi} \varphi_\sigma(\theta - \theta')(e^{-\varepsilon_K(\theta')} + e^{-\varepsilon_{\bar{K}}(\theta')})
$$
$$
+ \int \frac{d\theta'}{2\pi} \int_{\delta_\hbar}^1 d\sigma'\, s_{\max}\varphi_{\sigma,\sigma'}(\theta - \theta') \frac{e^{-(\sigma')^2\varepsilon_{\sigma'}(\theta')}}{(s_{\max}\sigma')^2} = 0\,, \tag{49}
$$

with the explicit cutoff $\delta_\hbar$, that we find self-consistently in what follows. In the integrals over the breathers, we single out the diverging part which can be then analytically computed by using Eq. (8)

$$
\int \frac{d\theta'}{2\pi} \int_{\delta_\hbar}^1 d\sigma'\, s_{\max}\varphi_{\sigma,\sigma'}(\theta - \theta') \frac{e^{-\sigma'^2\varepsilon_{\sigma'}(\theta')}}{(s_{\max}\sigma')^2} =
$$
$$
\int \frac{d\theta'}{2\pi} \int_{\delta_\hbar}^1 d\sigma'\, s_{\max}\varphi_{\sigma,\sigma'}(\theta - \theta') \frac{1}{(s_{\max}\sigma')^2} + \int \frac{d\theta'}{2\pi} \int_{\delta_\hbar}^1 d\sigma'\, s_{\max}\varphi_{\sigma,\sigma'}(\theta - \theta') \frac{e^{-\sigma'^2\varepsilon_{\sigma'}(\theta')} - 1}{(s_{\max}\sigma')^2} =
$$
$$
2\left[\log\left(\frac{\sigma}{\delta_\hbar}\right) + (1 - \sigma)\right] + \int \frac{d\theta'}{2\pi} \int_{\delta_\hbar}^1 d\sigma'\, s_{\max}\varphi_{\sigma,\sigma'}(\theta - \theta') \frac{e^{-\sigma'^2\varepsilon_{\sigma'}(\theta')} - 1}{(s_{\max}\sigma')^2}\,. \tag{50}
$$

Plugging this result back into the full equation (49) gives (we focus on the equation for the breathers, the equations for the kinks closely follow the same analysis)

$$
-\sigma^2\varepsilon_\sigma + 2\left[-\log\left(\hbar^{-1}\delta_\hbar s_{\max}\right) + 1 - \sigma\right] + \beta c^2 m_\sigma \cosh\theta
$$
$$
+ \int \frac{d\theta'}{2\pi} \varphi_\sigma(\theta - \theta')(e^{-\varepsilon_K(\theta')} + e^{-\varepsilon_{\bar{K}}(\theta')}) + \int \frac{d\theta'}{2\pi} \int_{\delta_\hbar}^1 d\sigma'\, \varphi_{\sigma,\sigma'}(\theta - \theta') \frac{e^{-(\sigma')^2\varepsilon_{\sigma'}(\theta')} - 1}{s_{\max}(\sigma')^2} = 0\,, \tag{51}
$$

where, indeed, the $\log\sigma$ singularities of the two terms exactly balance. One is left with the task of suitably choosing $\delta_\hbar$. To this end, we consider the $\sigma \to 0$ limit of this expression: if $\varepsilon_\sigma$ does not diverge, then $(e^{-(\sigma')^2\varepsilon_{\sigma'}(\theta')} - 1)/(\sigma')^2$ is non-singular and we can safely remove the cutoff in the integral. We now observe that the kernels vanish in this limit $\lim_{\sigma\to 0}\varphi_\sigma = \lim_{\sigma\to 0}\varphi_{\sigma,\sigma'} = 0$ as well as the breather mass $m_\sigma \to 0$. Therefore, taking the $\sigma \to 0$ limit of the equation within these assumptions, we are left with

$$
\lim_{\sigma\to 0}[\text{Eq. (51)}] \to \left[-\log\left(\hbar^{-1}\delta_\hbar s_{\max}\right) + 1 = 0\right]\,, \tag{52}
$$

which unambiguously fixes $\delta_\hbar$. A similar procedure can be carried out for the kink equations (48), leading to the finite expressions of the equations (27).

## C   The expectation value of the vertex operator

In this Appendix, we compute the expectation value of the vertex operator in the classical field theory by taking advantage of quantum results. To this end, one first notices that $\partial_m \hat{H} = \int \mathrm{d}x \frac{2mc^2}{g^2}(1 - \cos(g\phi))$, then uses the Hellmann-Ferynman theorem: for any eigenstate of the quantum Hamiltonian $|E\rangle$ it holds $\langle E|\partial_m \hat{H}|E\rangle = \partial_m(\langle E|\hat{H}|E\rangle) = \partial_m E$. The derivative of the energy is easy to compute, the only caveat is that one must derive the eigenvalues of finite-size eigenstates before eventually taking the thermodynamic limit. We notice that for each physical root density, there exist representative eigenstates $|E\rangle$ such that they are described by $\rho$ in the thermodynamic limit [30, 31]. These are standard computations in integrability [48, 69] leading to

$$
\frac{1}{L}\partial_m\langle H_q\rangle = \int \frac{\mathrm{d}\theta}{2\pi}\frac{M_q}{m}c\left(\cosh\theta\,\epsilon^{\mathrm{dr}}_{q;K}(\theta) - c\sinh\theta\,p^{\mathrm{dr}}_{q;K}\right)\frac{\rho_{q;K}(\theta)}{\rho^t_{q;K}(\theta)} +
$$
$$
\int \frac{\mathrm{d}\theta}{2\pi}\frac{M_q}{m}c\left(\cosh\theta\,\epsilon^{\mathrm{dr}}_{q;\bar{K}}(\theta) - c\sinh\theta\,p^{\mathrm{dr}}_{q;\bar{K}}\right)\frac{\rho_{q;\bar{K}}(\theta)}{\rho^t_{q;\bar{K}}(\theta)} +
$$
$$
\sum_n \int \frac{\mathrm{d}\theta}{2\pi}\frac{m_{q;n}}{m}c\left\{\cosh\theta\,\epsilon^{\mathrm{dr}}_{q;n}(\theta) - c\sinh\theta\,p^{\mathrm{dr}}_{q;n}(\theta)\right\}\frac{\rho_{q;n}(\theta)}{\rho^t_{q;n}(\theta)}. \tag{53}
$$

Above, $L$ is the system's size introduced for extensive reasons, and we have already approximated the quantum soliton mass with the semiclassical limit, hence $M_q = \hbar^{-1}M$, thus featuring a linear dependence in the bare mass $m$. Here, the superscript "dr" stands for the quantum dressing operation [29] analogous to the classical one in Eq. (20-21). For any test functions $\{\tau_K(\theta), \tau_{\bar{K}}(\theta), \tau_n(\theta)\}$, the quantum dressing operation $\{\tau_K(\theta), \tau_{\bar{K}}(\theta), \tau_n(\theta)\} \to \{\tau^{\mathrm{dr}}_K(\theta), \tau^{\mathrm{dr}}_{\bar{K}}(\theta), \tau^{\mathrm{dr}}_n(\theta)\}$ is defined by the following coupled integral equations

$$
\tau^{\mathrm{dr}}_{q;K}(\theta) = \tau_{q;K}(\theta) - \int \frac{\mathrm{d}\theta'}{2\pi}\varphi_q(\theta - \theta')\left(\tau^{\mathrm{dr}}_{q;K}(\theta')\frac{\rho_{q;K}(\theta')}{\rho^t_{q;K}(\theta')} + \tau^{\mathrm{dr}}_{q;\bar{K}}(\theta')\frac{\rho_{q;\bar{K}}(\theta')}{\rho^t_{q;\bar{K}}(\theta')}\right)
$$
$$
- \sum_n \int \frac{\mathrm{d}\theta'}{2\pi}\varphi_{q;n}(\theta - \theta')\tau^{\mathrm{dr}}_{q;n}(\theta')\frac{\rho_{q;n}(\theta')}{\rho^t_{q;n}(\theta')}, \tag{54}
$$

$$
\tau^{\mathrm{dr}}_{q;n}(\theta) = \tau_{q;n}(\theta) - \int \frac{\mathrm{d}\theta'}{2\pi}\varphi_{q;n}(\theta - \theta')\left(\tau^{\mathrm{dr}}_{q;K}(\theta')\frac{\rho_{q;K}(\theta')}{\rho^t_{q;K}(\theta')} + \tau^{\mathrm{dr}}_{q;\bar{K}}(\theta')\frac{\rho_{q;\bar{K}}(\theta')}{\rho^t_{q;\bar{K}}(\theta')}\right)
$$
$$
- \sum_{n'} \int \frac{\mathrm{d}\theta'}{2\pi}\varphi_{q;n,n'}(\theta - \theta')\tau^{\mathrm{dr}}_{q;n'}(\theta')\frac{\rho_{q;n'}(\theta')}{\rho^t_{q;n'}(\theta')}. \tag{55}
$$

From here, one can show how the classical dressing operation naturally emerges in the semiclassical limit. We first replace the quantum expressions with the classical ones with the correct scaling of $\hbar$ (see Eqs. (15,17, 19), pass over from the sum over breathers to an integral according to Eq. (14), and see that $\hbar$ drops out everywhere.

Note that through the non-singular parametrisation of the breather filling (28), an extra $(s_{\max}\sigma)^{-2}$ appears in the integral over the spectral parameter $\sigma$. Since this is inconvenient, we redefine the dressing operation for the breathers in such a way that $\tau^{\mathbf{dr}}_\sigma = \sigma^2 \left[\tau^{\mathrm{dr}}_n\right]_{\sigma = n\hbar/s_{\max}}$

while the kinks remain unaltered $\tau_K^{\mathbf{dr}} = \tau_K^{\mathrm{dr}}$. With this, we recover the dressing operation reported in the main text in Eq. (20-21). Using the quantum-classical correspondence for the dressing operation, the classical expectation value of the vertex operator (31) readily follows from Eq. (53).

# D    Numerical methods

In this Appendix we shortly overview the numerical methods used in this paper. A mathematica notebook for the Transfer Matrix approach used to compute equilibrium quantities (Figure 2) and a solver for the thermodynamic equation Eqs. (26-27) and partitioning protocols are available on Zenodo upon reasonable request [84]. However, we only provide the Monte Carlo data but not the source code, since it is a standard method.

## D.1    Solving the thermodynamics and partitioning

The first step is defining a convenient discretization of the phase space. To this end, we use a cartesian discretization that is nonlinear in the $\sigma$ space, building a tassellation of the domain phase space $[0, 1] \times [-\Lambda, \Lambda]$ in pairs $\{\sigma_i, \theta_i\}$, where $\Lambda$ is a large rapidity cutoff. In principle, also rapidities can be discretized according to a non-linear function, but this is not very important. In contrast, the non-linear discretization in $\sigma$ is chosen to be denser at the origin, i.e. where the Eqs. (26-27) become less regular. Each of these points is taken as a representative of a rectangle with edges placed on the midpoints between the chosen point and the neighbouring ones. The space of kinks is discretized only on the rapidities. The set of integral equations governing the thermodynamics are then discretized: in this respect, it is crucial to have a proper discretization of convolutions involving $\varphi$, which features singularities. Let us imagine $\varphi$ is convolved with a test function $\tau_\sigma(\theta)$ smooth in the phase space

$$\int \mathrm{d}\sigma \int \mathrm{d}\theta \varphi_{\sigma_i,\sigma}(\theta_j - \theta)\tau_\sigma(\theta) = \sum_{a,b} \int_{\frac{\sigma_a+\sigma_{a-1}}{2}}^{\frac{\sigma_a+\sigma_{a+1}}{2}} \mathrm{d}\sigma \int_{\frac{\theta_b+\theta_{b-1}}{2}}^{\frac{\theta_b+\theta_{b+1}}{2}} \mathrm{d}\theta \varphi_{\sigma_i,\sigma}(\theta_j - \theta)\tau_\sigma(\theta) \simeq$$

$$\sum_{a,b} \varphi^{(d)}_{\{\sigma_i,\theta_j\},\{\sigma_a,\theta_b\}} \tau_{\sigma_a}(\theta_b) \,, \tag{56}$$

$$\text{where} \quad \varphi^{(d)}_{\{\sigma_i,\theta_j\},\{\sigma_a,\theta_b\}} = \int_{\frac{\sigma_a+\sigma_{a-1}}{2}}^{\frac{\sigma_a+\sigma_{a+1}}{2}} \mathrm{d}\sigma \int_{\frac{\theta_b+\theta_{b-1}}{2}}^{\frac{\theta_b+\theta_{b+1}}{2}} \mathrm{d}\theta \varphi_{\sigma_i,\sigma}(\theta_j - \theta) \,.$$

We experienced that this discretization gives stable and fast-convergent results provided the integration kernel $\varphi^{(d)}_{\{\sigma_i,\theta_j\},\{\sigma_a,\theta_b\}}$ is well-approximated. To this end, we isolate the singular part $\varphi^{(S)}_{\sigma,\sigma'}(\theta)$ by defining it as

$$\varphi^{(S)}_{\sigma,\sigma'}(\theta) = \frac{2s_{\max}}{\pi} \log\left[\frac{\theta^2 + \frac{\pi^2}{4}(\sigma+\sigma')^2}{\theta^2 + \frac{\pi^2}{4}(\sigma-\sigma')^2}\right] + \frac{2s_{\max}}{\pi}\sigma\sigma' \log\left(\frac{\theta^2 + 1}{\theta^2 + \frac{\pi^2}{4}(\sigma+\sigma'-2)^2}\right) \,, \tag{57}$$

and the remaining being the non-singular part $\varphi^{(NS)}_{\sigma,\sigma'}(\theta) = \varphi_{\sigma,\sigma'}(\theta) - \varphi^{(S)}_{\sigma,\sigma'}(\theta)$. With this definition, $\varphi^{(S)}$ absorbs the singularities of $\varphi$ while retaining the same asymptotic behavior (it vanishes for large rapidities as well as if one of the two spectral parameters is sent to zero).

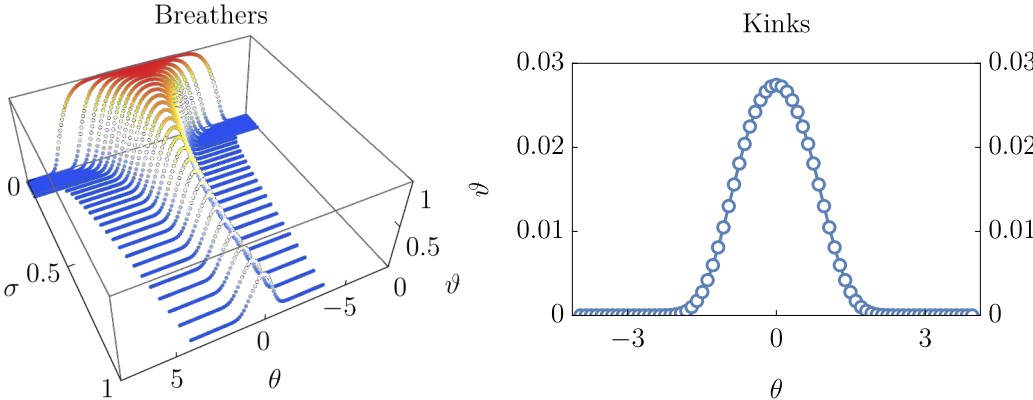

Figure 4: **The discretized solution of the filling function.** Here, we show the discretized filling function resulting from a numerical solution of the TBA equations. As an example, we show the case $g = m = c = 1$ and $\beta = 0.3$ and zero topological charge, using approximately 3000 points in total in the discretization. Left: filling function of breathers. Right: filling function of kinks (antikinks not reported, being identical to kinks) .

The singular part is simple enough to analytically perform the integral in Eq. (56), while the integral over the non-singular part is approximated as a constant over the integration domain

$$\varphi^{(d)}_{\{\sigma_i,\theta_j\},\{\sigma_a,\theta_b\}} \simeq \frac{\sigma_{a+1} - \sigma_{a-1}}{2} \frac{\theta_{b+1} - \theta_{b-1}}{2} \varphi^{(NS)}_{\sigma_i,\sigma_a}(\theta_j - \theta_b) + \int_{\frac{\sigma_a+\sigma_{a-1}}{2}}^{\frac{\sigma_a+\sigma_{a+1}}{2}} \mathrm{d}\sigma \int_{\frac{\theta_b+\theta_{b-1}}{2}}^{\frac{\theta_b+\theta_{b+1}}{2}} \mathrm{d}\theta \varphi^{(S)}_{\sigma_i,\sigma}(\theta_j - \theta) \,.$$

(58)

A similar discretization is employed for the kink-kink scattering shift and kink-breather one. With this approximation, the linear integral equations for the dressing are converted in linear matrix equations easy to numerically solve. The nonlinear equation determining the effective energy needs further care for a correct handling of the integral: let us focus only on this part for the sake of clarity and pick a term $\sigma_c$ from the spectral parameter discretization. Its role will become clear soon: we split the integral as follows

$$\int_{-\infty}^{\infty} \frac{\mathrm{d}\theta'}{2\pi} \int_0^1 \mathrm{d}\sigma' \, \varphi_{\sigma,\sigma'}(\theta - \theta') \frac{e^{-(\sigma')^2 \varepsilon_{\sigma'}(\theta')} - 1}{s_{\max}(\sigma')^2} =$$
$$\int_{-\infty}^{\infty} \frac{\mathrm{d}\theta'}{2\pi} \int_0^{(\sigma_c+\sigma_{c-1})/2} \mathrm{d}\sigma' \, \varphi_{\sigma,\sigma'}(\theta - \theta') \frac{e^{-(\sigma')^2 \varepsilon_{\sigma'}(\theta')} - 1}{s_{\max}(\sigma')^2} +$$
$$\int_{-\infty}^{\infty} \frac{\mathrm{d}\theta'}{2\pi} \int_{(\sigma_c+\sigma_{c-1})/2}^1 \mathrm{d}\sigma' \, \varphi_{\sigma,\sigma'}(\theta - \theta') \frac{e^{-(\sigma')^2 \varepsilon_{\sigma'}(\theta')}}{s_{\max}(\sigma')^2} +$$
$$\int_{-\infty}^{\infty} \frac{\mathrm{d}\theta'}{2\pi} \int_{(\sigma_c+\sigma_{c-1})/2}^1 \mathrm{d}\sigma' \, \varphi_{\sigma,\sigma'}(\theta - \theta') \frac{-1}{s_{\max}(\sigma')^2} \,.$$

For small values of the spectral parameter $\sigma'$, the kernel $\varphi_{\sigma,\sigma'}(\Delta\theta)$ is very peaked around small rapidity differences $\Delta\sigma \simeq 0$, but the support grows as $\sigma'$ gets larger. While $e^{-(\sigma')^2 \varepsilon_{\sigma'}(\theta')}$ quickly decays to zero for large rapidities (at fixed $\sigma'$), the $-1$ factor does not. By splitting the two terms we ensure that we can use a rapidity cutoff tailored on the fast decaying $e^{-(\sigma')^2 \varepsilon_{\sigma'}(\theta')}$. However, for small spectral parameters one wants to retain the two terms together, in order to balance the $1/(\sigma')^2$ singularity. The last line can be integrated exactly by using the fact that $\int d\theta \varphi_{\sigma,\sigma'}(\theta) = 4\pi s_{\max}\min(\sigma,\sigma')$, while the rest is discretized in the same spirit as Eq. (56) by reintroducing the cutoff in the rapidity space as well. The so-discretized nonlinear integral equations can be then solved with standard routines. The choice of $\sigma_c$ must be made in such a way that $\varphi_{\sigma,\sigma'<\sigma_c}(\Delta\theta)$ is very peaked in the rapidity space, having the smallest support as possible, while we still want to retain a few discretized points $\sigma_i < \sigma_c$ for a correct discretization of the integrals. Notice that since it holds by construction $\lim_{\sigma\to 1}\vartheta_\sigma(\theta) = 1$, the rapidity cutoff $\Lambda$ does not have to be necessarily chosen such that $\vartheta_\sigma(\theta)$ has support in $\lambda \in [-\Lambda, \Lambda]$ for every $\sigma$. It is rather sufficient requiring this only for $\sigma > \sigma_c$. Eventually, the convergence of the approximation upon increasing the cutoff and improving the discretization must be checked. As an example, in Fig. 4 we show the numerically computed filling function obtained with $\sim 3000$ points in the discretization.

## D.2 The Transfer Matrix approach

The transfer matrix approach is a standard method to convert one dimensional classical systems at equilibrium into zero-dimensional quantum mechanical problems, easy to be solved. Let us consider a Gibbs Ensemble on a finite size $[-L/2, L/2]$ with periodic boundary conditions, with the aim of computing the average of a given local obsevable of the phase field in $x = 0$, i.e. $O(\phi(0))$. Later, the observable can be chosen as the vertex operator. Within a path integral point of view

$$\langle O(\phi(0))\rangle = \frac{\int \mathcal{D}\phi\, O(\phi(0)) e^{\beta \int_{-L/2}^{L/2} dx \frac{1}{2}(\partial_x \phi)^2 + \frac{m^2 c^2}{g^2}(1-\cos(g\phi))}}{\int \mathcal{D}\phi\, e^{\beta \int_{-L/2}^{L/2} dx \frac{1}{2}(\partial_x \phi)^2 + \frac{m^2 c^2}{g^2}(1-\cos(g\phi))}} = \frac{\mathrm{Tr}[e^{-\frac{L}{2}\hat{H}_{\mathrm{eff}}} O(\phi) e^{-\frac{L}{2}\hat{H}_{\mathrm{eff}}}]}{\mathrm{Tr}[e^{-L\hat{H}_{\mathrm{eff}}}]}, \quad (59)$$

where the path integral is now seen as a propagator in imaginary time induced by a quantum mechanical thermal ensemble with effective temperature $L$, and an effective quantum Hamiltonian $\hat{H}_{\mathrm{eff}}$

$$\hat{H}_{\mathrm{eff}} = \frac{1}{2\beta}\partial_\phi^2 + \frac{m^2 c^2}{g^2}(1 - \cos(g\phi)). \quad (60)$$

In the thermodynamic limit $L \to +\infty$, the effective thermal ensemble is projected on the ground state. Therefore, $\langle O(\phi)\rangle$ is simply recovered by numerically computing the ground state of $\hat{H}_{\mathrm{eff}}$, and then taking the expectation value of the observable of interest. The only subtlety to be taken care of is that the potential is bounded and the field $\phi$ can get arbitrary large values. Nonetheless, we experienced that taking $\phi \in [-\pi n/g, \pi n/g]$ with $n$ a sufficiently large integer, imposing periodic boundary conditions and then discretizing the derivative operator over the interval converges for $n$ large enough. Notice that restricting to the first Brilluoin zone $n = 1$ is not sufficient.

## D.3 The Monte Carlo sampling

Monte Carlo simulations consist of two steps: *i)* a standard proposal-rejection scheme to sample the initial thermal distribution and *ii)* a deterministic evolution of the so-generated

field configuration with the equation of motion. The phase field is discretized on a equispaced grid of points with lattice spacing $a$, the classical Hamiltonian is likewise discretized: when sampling equilibrium ensembles, the distribution of the phase $\phi$ and the conjugated momentum factorize, the second being a simple i.i.d. Gaussian distribution for each lattice site. The $\phi-$part of the Hamiltonian is discretized as

$$H[\phi] = a \sum_j \frac{1}{2a^2}(\phi_{j+1} - \phi_j)^2 + \frac{m^2 c^2}{g^2}(1 - \cos(g\phi_j)). \tag{61}$$

Field configurations are then randomly updated by selecting a field site $j$ and proposing an update $\phi_j \to \phi'_j + \delta\phi$, with $\delta\phi$ Gaussianly distributed with zero mean. The new configuration is then accepted with probability $P = \exp(-\beta H[\phi'])/\exp(-\beta H[\phi])$. After a sufficient number of moves, the Markovian process samples the thermal distribution. To sample the partitioning protocols with two different temperatures and possibly topological charges, several choices can be made: for example, the inverse temperature $\beta$ can be promoted to be a smooth function interpolating between the two ensembles at the interface. However, we found it difficult to implement the topological charge imbalance in this setting. Therefore, we rather used Dirichlet boundary conditions by pinching the field in the center as well. More precisely, we enclose the field in two intervals $[-L/2, 0]$ and $[0, L/2]$ imposing these boundary conditions: at the center we fix $\phi(0) = 0$, while at the boundary $\phi(L/2)$ and $\phi(-L/2)$ are chosen according to the desired value of the topological charge (42), enforcing a microcanonical ensemble for the latter. In this setup, it is easy to account for two different temperatures as well. We stress that different ways to describe the interface at $t = 0$ will lead to different finite-time transient, but all of these will converge to the same late-time partitioning profile. For $t > 0$, the barrier is lifted and $\phi(0)$ is not pinned to zero, but rather allowed to evolve. In contrast, we retain Dirichlet boundary conditions at the two extrema of the system. The equations of motion are then discretized according to the scheme [57]

$$\phi_j(t+\mathrm{d}t) = 2\phi_j(t) - \phi_j(t-\mathrm{d}t) + \frac{\mathrm{d}t^2}{a^2}(\phi_{j+1}(t) + \phi_{j-1}(t) - 2\phi_j(t)) - \mathrm{d}t^2 \frac{m^2 c^2}{g} \sin(g\phi_j(t)), \tag{62}$$

which guarantees an all-time bounded discretization error, while other higher order methods such as Runge-Kutta would lead to exponential instabilities. We notice that simplectic methods exactly preserving integrability for any discretization order may be envisaged [58], but on the practical side we observed convergence for sufficiently small discretizations. The initial conditions for the above equations are obtained by assigning to $\phi_j(t = 0)$ a configuration sampled from the Monte Carlo, while $\phi(t = \mathrm{d}t) = \phi_j(t = 0) + \mathrm{d}t c^2 \Pi_j$, with $\Pi_j$ being Gaussianly distributed according to $\sim \exp(-\beta \frac{c^2}{2} \Pi_j)$.

In practice, for the parameters using in Figure 3 we attain convergence with a lattice spacing $a = 0.1$ and $\mathrm{d}t = 10^{-4}$. We use 4000 lattice sites for a total length $L = a \times 4000 = 400$, the field configurations are let to evolve until a maximum time $t = 110$ to avoid finite-size effects. For each parameter choice, we run 20 independent Monte Carlos, each of them collecting approximately 500 samples. Then, we consider the total average as the representative value and the errorbars are estimated as the variance over the independent runs.

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
