# Peer review of "Exact Thermodynamics and Transport in the Classical Sine-Gordon Model"

_SciPost Physics_

## Round 2 · Referee Report · Anonymous (Referee 1) · 2023-6-3

Strengths

1 - Timely topic
2 - Potentially relevant for experiments
3 - Clear presentation of the main argument

Weaknesses

1 - Some aspects need clarification
2 - The text needs some improvement regarding the use of English, and also eliminating typos.

See the requested changes.

Report

The work obtains the thermodynamical description and generalised hydrodynamics for sine-Gordon theory in the classical limit. This particular regime is important given the existence of experimental realisation exactly in this regime and therefore solves an important problem by modelling the non-equilibrium dynamics of the theory, with important potential applications to experiments. Consequently, the general acceptance criteria of Scipost Physics are satisfied.

Before the paper can be published, however, the authors should address some issues (c.f. the list of requested changes).

Requested changes

1 - Concerning the main idea, an obvious test would be to take the classical limit through the reflectionless points, where the explicit form of the TBA (and therefore the GHD) is known. This should be performed to back up all aspects of the present work, including the TBA, dressing equations and the analysis of the finite temperature expectation value of the cosine operator. In fact, I am quite surprised that the authors do not even comment on this aspect, as it does not seem to require much effort and could provide very strong support for their results.

2 - They write that the expectation value of the cosine operator can be readily taken in any Generalized Gibbs Ensemble. However, as far as I know, the GGE is not yet fully understood for the sine-Gordon model, despite some progress made in the work
E. Vernier and A.C. Cubero J. Stat. Mech. (2017) 023101. Therefore this statement seems quite superficial.

3 - The text would benefit very much from a thorough revision. A non-exhaustive list of examples is:
"a notorious integrable model" -> I do not think notorious is the right expression here
"far-fetching consequences" -> "far-reaching consequences"
"need to preliminary define" -> "need a preliminary definition of"
"is readily taken" -> "can be taken readily"
"the quantum scattering shift is readily extracted by the logarithmic derivative of the scattering matrix" -> "extracted by taking the"/ "extracted from the"/"extracted as the"
"one describes macrostates" -> "macrostates can be described"
"spatial extend" -> "spatial extension"
"so-obtained data" -> "resulting data"
"also rapidities can be discretized" -> "rapidities can also be discretized"
"a leap forward of the solitons" - this is called a time delay or trajectory shift
"since it holds by construction ..." -> "since ... holds by construction"
"likewise solitons, breathers" -> "similarly to solitons, breathers"
"as it will be unveiled" -> "as demonstrated later" / "as shown later"
"to rigorously take the thermodynamic limit" -> "to take the thermodynamic limit rigorously"
"take a different root" -> "take a different route"
"we converge to the correct result" -> "we arrive at the correct result"
"nowadays-standard" -> "by now standard"?

At some point the root density and total root density are called "two independent equations" - I believe they are rather "two independent quantities".

The charge (42) is conventionally called the topological charge. Also, what are "phase-slips"? I think the authors mean the winding number of the field.

There are also typos such as e.g. "Hyrdodynamics" - running a spell checker or a tool like Grammarly could help a lot in improving the text.

---

## Round 2 · Referee Report · Anonymous (Referee 2) · 2023-7-9

Strengths

1) Timely topic, experimentally relevant

2) clearly written (barring minor typos)

Weaknesses

1) Despite the main motivation being the experimental relevance of the semi-classical limit of the model, not much is mentioned about how the computed quantities relate to experiments.

2) The novelty of the analysis is not clear, the computation involves application of the well-known semiclassical limit.

3) Missing connections to old papers which first addressed these questions: Phys. Rev. B 26, 1430 (1982), Phys. Rev. B 25, 5806 (1982).

Report

The authors analyze the finite temperature properties of the semi-classical limit of the sine-Gordon model. While the topic is indeed timely, it is not immediately clear how this work answers a question that can in-fact be measured in experiments.

The authors start with the quantum model and take the semi-classical limit, regularizing the necessary divergences. It is not immediately clear if the computation is a straightforward application of well-known techniques or invents new techniques that could be used for other models.

Finally, the authors do not cite old papers on the topic [Phys. Rev. B 26, 1430 (1982), Phys. Rev. B 25, 5806 (1982)]. It will be good if the authors could address to what extent their computations add to what already exists in these works. Could the essential thermodynamic features be extracted from the beta^2 --> 0 limit (beta^2 = sine-Gordon coupling).

Requested changes

see report

---

## Editorial Decision

resubmitted